# On Testing for Biases in Peer Review

**Ivan Stelmakh, Nihar B. Shah and Aarti Singh**
School of Computer Science
Carnegie Mellon University
{stiv,nihars,aarti}@cs.cmu.edu

## Abstract

We consider the issue of biases in scholarly research, specifically, in peer review. There is a long standing debate on whether exposing author identities to reviewers induces biases against certain groups, and our focus is on designing tests to detect the presence of such biases. Our starting point is a remarkable recent work by Tomkins, Zhang and Heavlin which conducted a controlled, large-scale experiment to investigate existence of biases in the peer reviewing of the WSDM conference. We present two sets of results in this paper. The first set of results is negative, and pertains to the statistical tests and the experimental setup used in the work of Tomkins et al. We show that the test employed therein does not guarantee control over false alarm probability and under correlations between relevant variables, coupled with any of the following conditions, with high probability can declare a presence of bias when it is in fact absent: (a) measurement error, (b) model mismatch, (c) reviewer calibration. Moreover, we show that the setup of their experiment may itself inflate false alarm probability if (d) bidding is performed in non-blind manner or (e) popular reviewer assignment procedure is employed. Our second set of results is positive, in that we present a general framework for testing for biases in (single vs. double blind) peer review. We then present a hypothesis test with guaranteed control over false alarm probability and non-trivial power even under conditions (a)–(c). Conditions (d) and (e) are more fundamental problems that are tied to the experimental setup and not necessarily related to the test.

## 1 Introduction

Past research in social sciences indicates that humans display various biases including gender, race and age biases in many critical domains such as hiring [4], university admission [32], bail decisions [2] and many others. Our focus is on fairness in academia and scholarly research, and specifically, on biases in peer review. Peer review is a backbone of scholarly research and is employed by a vast majority of journals and conferences. Due to the widespread prevalence of the Matthew effect – rich get richer and poor get poorer – in academia [31, 27], any biases in peer review can have far reaching consequences on career trajectories of researchers. Specifically, we follow the long-standing debate [6, 25, 26, 1, 23, 8, 35, 14, and references therein] on whether the authors' identities should be hidden from reviewers or not. *The focus of this paper is on designing statistical tests to detect the presence of biases in peer review.*

In a recent remarkable piece of work, Tomkins et al. [33] conducted a large scale (semi-) randomized controlled trial during the peer review for the ACM International Conference on Web Search and Data Mining (WSDM) 2017. In their experiment, the entire pool of reviewers was partitioned uniformly at random into two equal groups – single blind and double blind – and each paper was assigned to two reviewers from each of the groups. In this manner, the peer-review data contained both single-blind and double-blind reviews for each paper. The experiment allowed them to conduct a causal inference to test for biases, and conclude that the single-blind system induces a bias in favor of papers authored by (i) researchers from top universities, (ii) researchers from top companies and (iii)

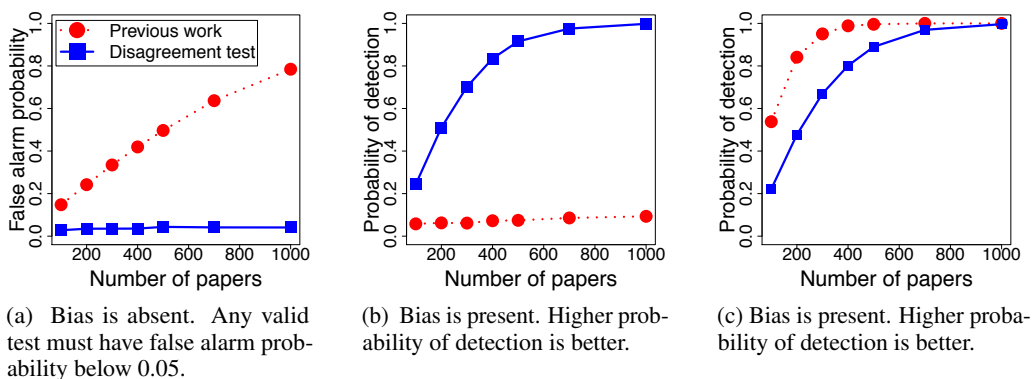

(a) Bias is absent. Any valid test must have false alarm probability below 0.05.

(b) Bias is present. Higher probability of detection is better.

(c) Bias is present. Higher probability of detection is better.

Figure 1: Synthetic simulations evaluating performance of the test in Tomkins et al. [33] ("previous work") and the test proposed in this paper ("DISAGREEMENT test"). Subfigures (a) and (b) are in presence of correlations and noisy estimates of true scores by double-blind reviewers; subfigure (c) has zero correlations and perfect estimate of true scores by double-blind reviewers. Details of the simulation setup are provided in Section 3. The error bars are too small to be visible.

famous authors. Interestingly, no bias against female-authored submissions was detected by their test, though a meta-analysis confirmed the presence of such bias. The conclusions of this experiment have had a significant impact. For instance, the WSDM conference itself completely switched to double-blind peer review starting 2018.

Testing for the presence of hypothesized phenomena is a common task in various branches of science including the biological, social, and physical sciences. The general approach therein is to impose a hard constraint on the probability of false alarm (claiming existence of the phenomenon when there is none; also called Type-I error) to some predefined threshold called significance level typically set as 0.05 or 0.01. The test would then aim to maximize the probability of detecting the phenomenon when it is actually present, while not violating the aforementioned hard constraint. The present paper also follows this general approach, for the specific setting of testing for biases using single versus double blind reviewing.

**Contributions.** In this paper, we study the problem of detecting bias in peer review (in the setting as considered in Tomkins et al. [33]). In this context, we present two sets of results.

**Negative results** (Section 3) We first analyze the testing procedure used by Tomkins et al., and show that under plausible conditions the statistical test employed therein does not control for false alarm probability. In other words, *we show that under reasonable conditions, the test used by Tomkins et al. [33] can, with probability as large as 0.5 or higher, declare the presence of a bias when the bias is in fact absent* (even when the test is tuned to have a false alarm error rate below 0.05). Specifically, we show that in presence of correlations that are reasonable to expect, any of the following factors breaks their false alarm probability guarantees: (a) measurement error caused by noise or subjectivity of reviewers, (b) model mismatch caused by violation of strong parametric assumptions on reviewers' behavior and (c) reviewer's calibration if she/he reviews more than one paper. Figures 1a and 1b illustrate the effect of measurement error on the false alarm probability and probability of detection of the test used by Tomkins et al. The issues we identify suggest that their test is at risk of committing Type-I error in declaring biases in their analysis.

Moving beyond the specific test used in Tomkins et al. [33], we also study the effect of their experimental design, which is simply the standard peer-review procedure with an additional random partition of reviewers into single and double blind groups. We show that two factors – (d) asymmetrical bidding procedure and (e) non-random assignment of papers to referees – as is common in peer-review procedures today may introduce spurious correlations in the data, breaking some key independence assumptions and thereby violating the requisite guarantees on testing.

**Positive results** (Sections 4 and 5) We propose a general framework for the design of statistical tests to detect biases in this problem setting, that overcomes the aforementioned limitations. Specifically, our

framework does not assume objectivity of reviewers and does not make any parametric assumptions on reviewers' behaviour. Conceptually, we propose to think of this problem as an instance of a two-sample testing problem where single-blind and double-blind reviews form two samples and the test operates on these samples. (In contrast, Tomkins et al. [33] study the problem under one-sample testing paradigm, operating on reviews of single-blind reviewers and using double-blind reviews to estimate some parameters in their parametric model).

We then design a computationally-efficient hypothesis test with a provable control over the false alarm probability under various conditions, including aforementioned conditions (a) - (c). Our test also has non-trivial power in that it has considerably higher probability of detection in hard cases where the test used by Tomkins et al. fails, and also has not too much loss in power when the assumptions made in Tomkins et al. [33] are exactly met, and there is no correlation or noise. The performance of this test is illustrated in Figure 1.

It is important to note that in this work, we do not aim to prove or disprove the existence of biases declared in the experiment by Tomkins et al. [33]. Instead, our focus is on the theoretical validity of the statistical procedures used to conduct such experiments and more generally on principled statistical approach towards designing such experiments. Finally, we note that the results and tests we discuss in this work are also applicable beyond peer review, and can be used to test for biases in other domains such as admissions and hiring.

**Related work.** The problem of identifying biases in human decisions is commonly studied in social science and there are many works that design and conduct randomized field experiments in various settings, including resume screening [5], hiring in academia [22], and peer review [6, 23]. However, the conference peer review setup we consider in this work does not comprise a fully randomized control trial (i.e., the reviewers are not assigned to submissions at random) and past approaches fail due to idiosyncrasies of the peer-review process. For example, a popular approach [5, 22] is to assign author identities to (fabricated) documents (resumes, application packages or papers) uniformly at random and compare the outcomes for different categories of authors. In our setup, *random assignment of author identities to real (i.e., non-fabricated) submissions* is problematic due to various logistical and ethical issues such as reviewers guessing actual authors thereby causing biases, and requirements of getting authors to agree to have their paper/name modified. Another approach [23] is to submit *the same paper* to multiple reviewers in both single-blind and double-blind conditions and test for the difference in the acceptance rates between conditions. However, such an approach necessitates a considerable additional reviewing load. Other approaches include observational studies, and we refer the interested readers to [33] for a more in-depth literature review.

This paper also falls in the line of several recent works in computer science on the peer-review process which includes both empirical [15, 13] and theoretical [30, 34, 17] studies.

## 2 Preliminaries

The general peer-review setup we study for testing biases using single and double blind review is as considered in Tomkins et al. [33]. We study a conference peer-review setup where $n$ papers are submitted at once and $m$ independent reviewers are available to review submissions. With a goal to test whether single-blind reviewing induces a bias against or in favor of some groups of authors, we consider some pre-defined set of $k$ binary mutually non-exclusive properties pertaining to the author(s) of any paper to be tested for bias. For example, a property could be "the first author is female" or "majority of authors are from the USA". Each paper $j \in [n]$ is then associated with $k$ indicator variables $w_j^{(1)}, \ldots, w_j^{(k)}$, where $w_j^{(\ell)} = 1$ if paper $j$ satisfies property $\ell$ and $w_j^{(\ell)} = -1$ otherwise. For each $\ell \in [k]$ we let $\mathcal{J}_\ell$ denote the set of papers that satisfy property $\ell$ and $\overline{\mathcal{J}}_\ell = [n] \backslash \mathcal{J}_\ell$ denote its complement.[1]

The peer review process is conducted as follows. Each reviewer is uniformly at random allocated to one of the two conditions: (i) Double-Blind condition (DB) in which reviewers do not observe identities of papers' authors; and (ii) Single-Blind condition (SB) in which reviewers observe identities of papers' authors. Next, each paper is assigned to $\lambda$ reviewers from the SB group and $\lambda$ reviewers from the DB group such that each reviewer reviews at most $\mu$ submissions, where $\lambda$ and $\mu$ are predefined constants. In both conditions, if any reviewer $i \in [m]$ is assigned to any paper $j \in [n]$,

then she/he returns a binary accept/reject recommendation and possibly a numeric score that estimates a quality of the paper as perceived by the reviewer, accompanied by a textual review.

For each property $\ell \in [k]$, we are interested in whether *single-blind peer review setup induces a bias* against or in favor of papers that satisfy this property. For example, if we consider property "the first author is female", then we aim at testing for the bias against or in favor of papers with female first author. Note that with respect to the properties, the study is observational in that we cannot assign author identities to papers at random. Hence, the effect of confounding is unavoidable and utmost care must be taken to address presence of confounding factors.

For brevity, in this paper we consider the case of a single property of interest ($k = 1$) which captures the complexity of our problem. For ease of notation, we drop index $\ell$ from $w^{(\ell)}$ and $\mathcal{J}_\ell$. For the discussion of the general case of multiple properties of interest ($k > 1$) we refer the reader to the extended version of this paper [29].

Let us now give details of the testing procedure used by Tomkins et al. [33].

**Model and test used by Tomkins et al.** We begin by introducing an idealized version of their model. They assume a parametric, logistic model for the binary decisions made by SB reviewers. Specifically, for each paper $j \in [n]$, let $Y_{1j}, \ldots, Y_{\lambda j}$ denote the binary accept/reject decisions given by the $\lambda$ reviewers assigned to paper $j$ in the SB setup. It is assumed that $\{Y_{rj}\}_{r \in [\lambda]}$ are independent draws from a Bernoulli random variable with an expectation $\pi_j$ satisfying

$$\log \frac{\pi_j}{1 - \pi_j} = \beta_0 + \beta_1 q_j^* + \beta_2 w_j, \tag{1}$$

where $q_j^*$ is a "true" underlying score of paper $j$, $w_j$ is an indicator of property satisfaction and $\{\beta_0, \beta_1, \beta_2\}$ are unknown coefficients. In words, the model says that if there is a positive (negative) bias with respect to a property of interest, then the fact that paper satisfies the property increases (decreases) the log-odds of the probability of recommending acceptance by $2\beta_2$ as compared to the case if the same paper does not satisfy the property. The main difficulty with this model in the peer review setting lies in the fact that true scores $\{q_j^*, j \in [n]\}$ are unknown and hence standard tests for logistic regression model are not readily applicable.

In order to overcome the unavailability of true scores $\{q_j^*, j \in [n]\}$ in the model (1), Tomkins et al. [33] use a plug-in estimate: they replace $q_j^*$ with the mean $\widetilde{q}_j$ of scores given by the DB reviewers to paper $j$, for every $j \in [n]$. Under this approximation and using $\widetilde{q}_1, \ldots, \widetilde{q}_n$, they obtain maximum likelihood estimates of coefficients $\{\widehat{\beta}_0, \widehat{\beta}_1, \widehat{\beta}_2\}$ and then use the standard Wald test [36] to test for significance of the coefficient $\beta_2$. A bias is declared present if the coefficient $\beta_2$ is found significant; the direction of the bias is determined as the sign of $\widehat{\beta}_2$.

## 3  Negative results

In this section we identify several issues that should be taken into account when testing for biases in the setup we consider. Noting that the issues themselves are general, we motivate and discuss them in context of the prior work by Tomkins et al. [33] and investigate possible consequences of these issues through synthetic simulations.

Recall that with respect to the property of interest the experiment is observational and hence we cannot assume that variable $w$ that encodes property satisfaction is independent of true score $q^*$. For example, consider a property "paper has author from top univeristy". For this property a non-trivial correlation between true scores and indicators of property satisfaction is natural to expect. While correlation itself does not cause issues, we identify five conditions that coupled with correlation can be significantly harmful.

In the simulations to follow, we juxtapose the algorithm by Tomkins et al. [33] to our DISAGREEMENT test introduced later in the paper. Complete details of all simulations are given in Appendix A.

**(a) Measurement error.** Tomkins et al. [33] report low interreviewer agreement between DB reviewers which means that the estimates $\widetilde{q}_1, \ldots, \widetilde{q}_n$ of the true scores by the DB reviewers are noisy. It is known [28, 7] that noisy covariate measurement coupled with correlation between some covariates may inflate Type-I error rate of the Wald test for logistic regression. We now investigate the impact of measurement error on the Type-I error rate of the Tomkins et al. test through simulations.

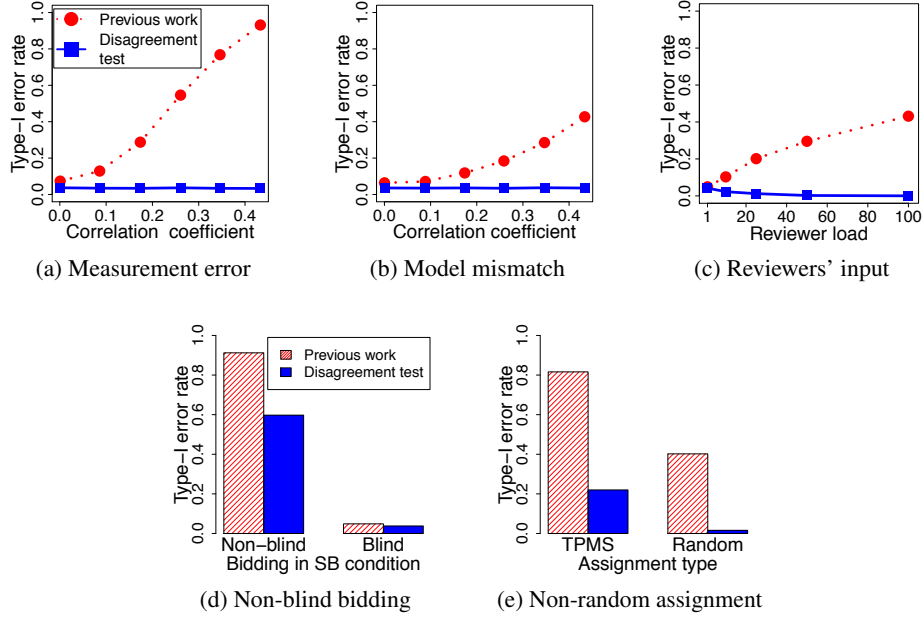

(a) Measurement error  (b) Model mismatch  (c) Reviewers' input

(d) Non-blind bidding  (e) Non-random assignment

Figure 2: Type-I error of the test from previous work (Tomkins et al. [33]) blows up under five different setups: bias is absent in all simulations and the tests are designed to limit the Type-I error to at most 0.05. In contrast, our DISAGREEMENT test is robust to violations of modelling assumptions (a)-(c). Note that non-blind bidding by SB reviewers and non-randomness of the assignment (left bars in subfigures d and e), which pertain to the experimental setup rather than the modelling, break guarantees of both tests. Error bars are too small to be visible.

We consider absence of any bias, and assume that model (1) with $\beta_2 = 0$ is correct for both DB and SB reviewers. We consider DB reviewers to report noisy estimates of true scores $q_j^*$, and vary the correlation between $q^*$ and $w$. We plot the Type-I error rates in Figure 2a for the test in Tomkins et al. [33] and our proposed test; both tests are designed to restrict the Type-I error rate to 0.05. Given that interreviewer agreement in the actual experiment of Tomkins et al. [33] was low (level of noise is high), the fact that some properties they consider may lead to correlations between $q^*$ and $w$ is concerning, because it could potentially undermine the validity of their findings.

The simulations in Section 1 follow the setup presented here: Figures 1a and 1b consider measurement error with correlation fixed at 0.4 and 0.6 respectively. Notice that issues exacerbate as sample size grows; Figure 1c has zero correlation and no measurement errors.

**(b) Model mismatch.** Model (1) assumes a specific parametric relationship, which is unlikely to hold in practice. In order to check the effect of model mismatches, we consider a violation of the model (1) and suppose that the correct model is $\log \frac{\pi_j}{1-\pi_j} = \beta_0 + \beta_1 \left(q_j^*\right)^3 + \beta_2 w_j$. We consider an absence of any bias, that is, set $\beta_2 = 0$ for both SB and DB reviewers. We perform simulations similar to those in item (a) with the exception that true scores $q_j^*, j \in [n]$, are known exactly to the test of Tomkins et al. Figure 2b shows results of simulations.

**(c) Reviewer calibration.** Model (1) assumes that reviews given by the same reviewer are independent. In practice this assumption may be violated due to correlations introduced by reviewer's calibration [34]. While some easy calibrations such as harshness/leniency can be captured by simple parametric extensions of model (1), more subtle patterns are beyond the scope of this model. Suppose for example that the strength of reviewers' input depends on paper's clarity – the better the paper written, the lower the contribution due to reviewers' calibration. Assume also that we are given a set of papers such that true score of each paper is proportional to the clarity of the paper (we formalize construction in Appendix A.1.3). Coupled with the correlation between $q^*$ and $w$, this pattern is sufficient to break Type-I error guarantees of Tomkins et al. test. Figure 2c shows a result of simulations in which we vary the number of papers per reviewer, keeping correlation between $q^*$

and $w$ fixed at 0.75 and the total number of papers fixed at $n = 1000$. We simulate a wide range of reviewer load $\mu$ including small to medium loads of 5-15 papers typical in machine learning conferences like NeurIPS and larger loads of 40 or higher found in other smaller conferences.

**(d) Non-blind bidding.** The issues discussed above pertain to the testing procedure and modelling assumptions made by Tomkins et al. [33], and can be avoided by designing a principled statistical approach to the testing problem as we do in Sections 4 and 5. We now issue a commentary regarding the experimental setup considered in their work and show that the setup itself may create problems in controlling the Type-I error. In the experiment of Tomkins et al., papers are allocated to reviewers based on "bids" representing their preferences. Importantly, the reviewers in the SB setup also get to see author identities in the bidding stage, which may act as a confounding factor in tests for bias in the acceptance/rejection of papers. Indeed, authorship information available to SB reviewers may introduce a difference in bidding behaviour between conditions and this difference may result in structurally different evaluations even when reviewers are unbiased, leading to a blow-up of the Type-I error rate of any reasonable test as illustrated by Figure 2d (formal setup is in Appendix A.1.4). This issue is indeed pointed out as a caveat by Tomkins et al. in their paper.

**(e) Reviewer assignment.** Previous issue is indeed pointed out as a caveat by Tomkins et al. in their paper. One might imagine that a natural solution to the aforementioned problem would be to conduct the bidding in a double blind fashion for both groups. However, perhaps surprisingly, we show that even if both groups bid in a double blind fashion (or even if the bidding process is eliminated entirely), and even if the reviewers are assigned to DB or SB groups uniformly at random, the non-random assignment using algorithms such as TPMS [9] can still swell the Type-I error rate. Figure 2e contrasts Type-I errors under random assignment of reviewers to papers, and an assignment computed by the TPMS algorithm [9] operating on a similarity matrix constructed in Appendix A.1.5, where reviewer decisions are correlated with the similarity (e.g., reviewers being more lenient on papers closer to their own area). Notably, even the DISAGREEMENT test which is robust to various issues discussed above is unable to control the Type-I error under the TPMS assignment. Due to measurement errors that arise in our construction, the test of Tomkins et al. [33] fails even under random assignment, and the non-randomness exacerbates the effect.

## 4 Novel framework to test for biases

In Section 3 we identified five key limitations of the approach taken by Tomkins et al. [33]. Three of these limitations pertain to the testing procedure and the other two limitations regarding the bidding and assignment procedures relate to the design of experiment itself. In the next two sections we aim at improving the test used by Tomkins et al. [33]. Hence, in the theoretical arguments below we assume that reviewers' evaluations are independent of bids and similarities (alternatively, the assignment of papers to referees is selected uniformly at random from the set of all feasible assignments). In the extended version of this paper [29] we relax the random assignment assumption by introducing a novel experimental procedure that allows to use any assignment algorithm without running into issues (d) and (e) discussed above.

One approach to address the aforementioned issues with the testing procedure is to design methods for logistic regression model that are robust to various factors such as noise, misspecification, etc. [20, 18, 28, 24, 7]. However, in this work we consider the problem more generally, because the logistic model (1) itself could be highly inaccurate. Specifically, we aim at designing a test that does not rely on strong modelling assumptions and also holds when reviewers decisions are subjective.

At a high level, our approach to testing for biases is different from those proposed by Tomkins et al. [33] in two ways. First, we relax two strict modelling assumptions: (i) instead of assuming existence of true qualities of submissions, we allow subjectivity in reviewer evaluations [16, 11, 3, 21, 19], and (ii) we do not assume any specific form of the relationship between a paper and its probability of acceptance by a reviewer. Instead, we allow these probabilities to be completely arbitrary and define bias in terms of these probabilities. Second, we treat this problem conceptually differently from the work of Tomkins et al. [33]. The test therein treats the problem as that of one-sample testing and uses DB scores as a plugin estimate of true scores in SB model. In contrast, we approach this problem through the lenses of two-sample testing, where SB and DB reviews form the two samples, and the goal is to test whether they belong to the same distribution. This perspective helps us to avoid a number of issues discussed in Section 3.

Formally, let $\Pi^{\text{db}} \in [0,1]^{m \times n}$ be a matrix whose $(i,j)^{\text{th}}$ entry, denoted as $\pi_{ij}^{(\text{db})}$, represents a probability that reviewer $i$ would recommend acceptance of paper $j$ if that paper is assigned to that reviewer in DB setup. Similarly, let matrix $\Pi^{\text{sb}} \in [0,1]^{m \times n}$ be an analogous matrix in SB setup, and denote its $(i,j)^{\text{th}}$ entry as $\pi_{ij}^{(\text{sb})}$.

Let $\mathcal{R}_{\text{SB}}$ be the set of reviewers allocated to the SB condition. Moreover, for each $i \in \mathcal{R}_{\text{SB}}$, let $\mathcal{P}_{\text{SB}}(i)$ denote the set of papers assigned to reviewer $i$ and let $Y_{ij} \in \{0,1\}$ denote the accept/reject decision given by reviewer $i$ for paper $j \in \mathcal{P}_{\text{SB}}(i)$. We similarly define the set of DB reviewers $\mathcal{R}_{\text{DB}}$ and their decisions $\{X_{ij} : i \in \mathcal{R}_{\text{DB}}, j \in \mathcal{P}_{\text{DB}}(i)\}$. We are interested in testing for biases with respect to a property of interest. Recalling our notation $\mathcal{J} \subseteq [n]$ for the set of papers that satisfy a property of interest, and $\overline{\mathcal{J}}$ as its complement, we now define the bias testing problem.

**Problem 1** (Bias testing problem). Given significance level $\alpha \in (0,1)$, and decisions of SB and DB reviewers, the goal is to test the following hypotheses:

$$H_0 : \forall i \in [m] \ \forall j \in [n] \ \ \pi_{ij}^{(\text{sb})} = \pi_{ij}^{(\text{db})}$$

$$H_1 : \forall i \in [m] \ \forall j \in [n] \ \begin{cases} \pi_{ij}^{(\text{sb})} \geq \pi_{ij}^{(\text{db})} & \text{if } j \in \mathcal{J} \\ \pi_{ij}^{(\text{sb})} \leq \pi_{ij}^{(\text{db})} & \text{if } j \in \overline{\mathcal{J}}, \end{cases} \qquad (2)$$

where at least one inequality in the alternative hypothesis (2) is strict.[2]

In words, under the null hypothesis the knowledge of authors' identities does not induce any difference in reviewers' behaviour. On the other hand, under the alternative there is a bias in favor of papers that satisfy the property of interest. Note that one can define an alternative that represents a bias against papers from $\mathcal{J}$ simply by exchanging the sets $\mathcal{J}$ and $\overline{\mathcal{J}}$ in (2).

Our goal is to design a testing procedure that both controls for the Type-I error and has non-trivial power for any pair of matrices $\Pi^{\text{sb}}, \Pi^{\text{db}}$ that fall under the definition of Problem 1.

**Non-trivial power.** Informally, we say that the test has non-trivial power if for choices of $\Pi^{\text{sb}}$ and $\Pi^{\text{db}}$ for which the presence of bias is "obvious", the test is able to detect the bias with probability that goes to 1 as number of papers in both $\mathcal{J}$ and $\overline{\mathcal{J}}$ grows to infinity. Formally, we say that matrices $\Pi^{\text{sb}}$ and $\Pi^{\text{db}}$ satisfy the alternative hypothesis (2) with margin $\delta$, if all inequalities in equation (2) are satisfied with margin $\delta > 0$, that is, $|\pi_{ij}^{(\text{sb})} - \pi_{ij}^{(\text{db})}| > \delta \ \forall (i,j) \in [m] \times [n]$. Then we say that the testing procedure has non-trivial power if for any $\varepsilon > 0$ and for any $\delta > 0$ there exists $n_0 = n_0(\varepsilon, \delta)$ such that if $\min\{|\mathcal{J}|, |\overline{\mathcal{J}}|\} > n_0$, then for any $\Pi^{\text{sb}}$ and $\Pi^{\text{db}}$ that satisfy alternative hypothesis (2) with margin $\delta$, the power of testing procedure is at least $1 - \varepsilon$.

For instance, if the logistic model (1) is correct for both SB and DB reviewers with $\beta_0^{(\text{sb})} = \beta_0^{(\text{db})} = \beta_0$, $\beta_1^{(\text{sb})} = \beta_1^{(\text{db})} = \beta_1$, $\beta_2^{(\text{db})} = 0$ and $|\beta_2^{(\text{sb})}| > 0$, then the requirement of non-trivial power ensures that for any choice of true scores bounded in absolute value by a universal constant and any choice of property satisfaction indicators, the test has power growing to 1 as $\min\{|\mathcal{J}|, |\overline{\mathcal{J}}|\}$ goes to infinity.

## 5 Positive results

In this section we present a test for Problem 1 and discuss some generalizations.

### 5.1 Disagreement-based test

Let us now introduce a statistical test for bias testing problem (Problem 1) and show that it satisfies requirements of control over Type-I error and has non-trivial power. The test is built on two key ideas.

- First, consider a pair of SB and DB reviewers who disagree in their decisions for some paper. Under the null hypothesis, the events "SB accepts and DB rejects" and "SB rejects and DB accepts" are equiprobable. In contrast, under the alternative, SB reviewer is more (or less) likely to vote for acceptance than her/his DB counterpart, depending on the value of $w_j$ and the direction of the bias.

- Second, in order to avoid correlations introduced by reviews given by the same reviewer, the test uses at most one decision per reviewer. It does so by first matching reviewers into pairs, consisting of one SB and one DB reviewer who review a common paper, and maximizing the number of such pairs subject to a constraint that each reviewer appears in at most one pair.

For a moment, assume that we are given a set of tuples $\mathcal{T}$, where each tuple $t \in \mathcal{T}$ consists of a paper $j_t \in [n]$, decision of a SB reviewer for this paper $Y_{j_t}$, decision of a DB reviewer for this paper $X_{j_t}$ and indicator of property satisfaction $w_{j_t}$, with a constraint that each reviewer contributes her/his decision to at most one tuple. In this setting, we formally describe our DISAGREEMENT test as Test 1.

---

**Test 1** DISAGREEMENT

**Input:** Significance level $\alpha \in (0, 1)$.
    Set of tuples $\mathcal{T}$, where each $t \in \mathcal{T}$ is of the form $(j_t, Y_{j_t}, X_{j_t}, w_{j_t})$ for some paper $j \in [n]$.
1. Initialize $U$ and $V$ to be empty arrays.

2. For each tuple $t \in \mathcal{T}$, if $Y_{j_t} \neq X_{j_t}$, append $Y_{j_t}$ to $\begin{cases} U & \text{if } w_{j_t} = 1 \\ V & \text{if } w_{j_t} = -1 \end{cases}$.

3. Run permutation test [12] at the level $\alpha$ to test if entries of $U$ and $V$ are exchangeable random variables, using the test statistic:

$$\tau = \frac{1}{|U|} \sum_{r \in [|U|]} U_r - \frac{1}{|V|} \sum_{r \in [|V|]} V_r.$$

4. Reject the null if and only if the permutation test rejects the null. (If any of the arrays $V$ and $U$ are empty, the test keeps the null.)

---

We now discuss construction of the set $\mathcal{T}$ for input to Test 1 from the given set of reviews. The goal of the construction is to ensure that $\mathcal{T}$ contains "enough" tuples that correspond to papers from $\mathcal{J}$ and $\overline{\mathcal{J}}$. We consider two cases.

- If $\lambda \geq \mu$, then using the Hungarian matching algorithm each paper is matched to 1 SB reviewer and 1 DB reviewer, in a manner that each reviewer is matched to at most one paper.
- If $\lambda < \mu$, then we use an iterative algorithm which greedily matches one paper from $\mathcal{J}$ and one paper from $\overline{\mathcal{J}}$ to 1 SB and 1 DB reviewer in each iteration, with a constraint that each reviewer is matched to at most one paper. This algorithm is guaranteed to match a constant fraction of papers from both $\mathcal{J}$ and $\overline{\mathcal{J}}$.

The matching algorithms for both cases are formally presented in Appendix C due to lack of space. The following theorem now presents guarantees for our test.

**Theorem 1.** *For any significance level $\alpha \in (0, 1)$, under the setup of the bias testing problem (Problem 1), the DISAGREEMENT test coupled with matching algorithms from Appendix C is guaranteed to control for the Type-I error at the level $\alpha$, and also satisfy the requirement of non-trivial power.*

**Remark.** If the logistic model (1) is correct, then Theorem 1 ensures that the DISAGREEMENT test provably controls the Type I error and can detect a bias with probability that goes to 1 as sample size grows, without requiring knowledge (neither exact nor approximate) of true scores $q_1^*, \ldots, q_n^*$.

We now discuss the issues (a)-(c) discussed in Section 3 in the context of our DISAGREEMENT test.

- **Measurement error.** Our test does not rely on any estimation of papers' qualities made by reviewers. Moreover, we do not even assume that there exists some objective quantity that can be estimated. Hence, our test does not suffer from issues caused by noisy estimates of scores given by DB reviewers as illustrated by Figure 2a.
- **Model mismatch.** The only assumption we make is that under the null hypothesis there is no difference in behavior of SB and DB reviewers. Hence, Proposition 1 guarantees that our test is robust to violations of specific parametric model (1) as illustrated by Figure 2b.
- **Reviewer calibration.** We circumvent the detrimental effect of spurious correlations introduced by reviewers' calibration through a matching procedure that ensures that each reviewer contributes at most one review to the test. See Figure 2c for an illustration. Of course, such robustness comes at the cost of some power, but we notice that our matching procedures guarantee use of at least a constant fraction of available data, thereby limiting the reduction in the power.

**Effect size.** The test statistic $\tau$ of the DISAGREEMENT test gives an estimate of the effect size. Slightly informally, $\tau$ measures the difference in acceptance rates of "borderline" papers from $\mathcal{J}$ and $\overline{\mathcal{J}}$ in the SB setup. Indeed, by conditioning on pairs of disagreeing reviewers in Step 2 of Test 1, the test rules out "clear accept" and "clear reject" papers thus considering only the papers for which

reviewers disagree (i.e. borderline papers). The absolute value of the test statistic then is a reasonable estimate of the effect size and is in a similar vein to Cohen's $d$ [10] and other popular measures.

## 5.2 Generalization

We now consider a generalization of Problem 1 which accommodates an additional confounding factor — a bias in the reviewer simply due to her/his assignment in the SB or the DB group (and independent of the paper or its characteristics). For example, reviewers may not have any bias with respect to the property of interest, but just being placed in the SB condition may induce more harsh opinions from the reviewers in DB. Formally, recall the null hypothesis $\pi_{ij}^{(\text{sb})} = \pi_{ij}^{(\text{db})} \; \forall (i,j) \in [m] \times [n]$ in Problem 1. Instead, under the null, we now allow $\pi_{ij}^{(\text{sb})} = f_0(\pi_{ij}^{(\text{db})})$, for some monotonic function $f_0 : [0,1] \to [0,1]$. As in Problem 1, the bias is then defined as a violation of the null hypothesis where direction of the violation is determined by the indicator $w_j$.

Of course, one may not know the function $f_0$ and the goal of this general problem is to design a test that is guaranteed to control over the Type-I error and has non-trivial power uniformly for all functions $f_0$ that belong to some set of functions $\mathcal{F}^*$. The definition of non-trivial power from Section 4 transfers to this problem with the exception that all $\pi_{ij}^{(\text{db})}$ are substituted by $f_0(\pi_{ij}^{(\text{db})})$ for $f_0 \in \mathcal{F}^*$. In the extended version of this paper [29] we present a negative result showing that this goal is impossible to achieve for general $\mathcal{F}^*$. However, in what follows we show that one can achieve this goal under some specific choices of class $\mathcal{F}^*$, including a generalization of logistic model (1).

**Generalized logistic model.** For every paper $j \in [n]$, let $q_j \in \mathbb{R}$ be some unknown representation of paper $j$. The generalized logistic model assumes that for every $(i,j) \in [m] \times [n]$ we have

$$\text{DB: } \log \frac{\pi_{ij}^{(\text{db})}}{1 - \pi_{ij}^{(\text{db})}} = \beta_0^{(\text{db})} + \beta_1^{(\text{db})} q_j, \qquad \text{SB: } \log \frac{\pi_{ij}^{(\text{sb})}}{1 - \pi_{ij}^{(\text{sb})}} = \beta_0^{(\text{sb})} + \beta_1^{(\text{sb})} q_j + \beta_2^{(\text{sb})} w_j, \qquad (3)$$

where $q_j, j \in [n]$, and coefficients are bounded in absolute value by constant $B$. The goal under this model is to test whether $\beta_2^{(\text{sb})} = 0$. The model is called "generalized", because it does not assume that $q_j$ has a known meaning or that it can be measured. For instance, it may be that $q_j = q_j^*$, or that $q_j = (q_j^*)^3$, or $q_j$ may be a complex function of the content of the paper. The generalized logistic model (3) falls in the framework of Problem 1 if $\beta_0^{(\text{db})} = \beta_0^{(\text{sb})}$ and $\beta_1^{(\text{db})} = \beta_1^{(\text{sb})}$. Having defined necessary terminology, we are now ready to formulate the main result of this section.

**Theorem 2.** *For any significance level $\alpha \in (0,1)$, suppose that the generalized logistic model (3) is correct. If $\beta_1^{(sb)} = \beta_1^{(db)}$, then the DISAGREEMENT test is guaranteed to keep the Type-I error below $\alpha$, and also satisfy the requirement of non-trivial power irrespective of whether $\beta_0^{(db)} = \beta_0^{(sb)}$ or $\beta_0^{(db)} \neq \beta_0^{(sb)}$. Conversely, if we allow both $\beta_0^{(sb)} \neq \beta_0^{(db)}$ and $\beta_1^{(sb)} \neq \beta_1^{(db)}$, then no test that operates on decisions of SB and DB reviewers can control for the Type-I error and simultaneously satisfy the requirement of non-trivial power.*

Theorem 2 shows that the generalized bias testing problem is much harder than the original Problem 1 as there exists no algorithm that solves this problem in full generality even under specific model (3).

## 6 Discussion

In this work we consider the problem of testing for biases in peer review. We show that under various conditions the approach used by prior work does not control over the Type-I error rate. We underscore that we do not aim at disproving the presence of biases found in the past work, but our focus is on validity of testing methods. With this goal in mind, we propose a principled approach towards testing for biases in peer review and design a test that provably controls for the Type-I error rate and also satisfy the requirement of non-trivial power under minimal assumptions. As we showed in Section 5.2 (more detailed discussion can be found in the extended version of this paper [29]), in general the assumptions we make cannot be relaxed without sacrificing the non-trivial power requirement or control over the Type-I error. On a separate note, we also demonstrate that the experimental setup of Tomkins et al. [33], which uses standard procedures for (non-random) assignment of papers to reviewers, can itself break Type-I error guarantees of statistical procedures. In the extended version of this paper [29] we design a novel experimental procedure which (i) is amenable to standard conference peer review procedures, and (ii) does not violate Type-I error guarantees of our DISAGREEMENT test.

## Acknowledgments

This work was supported in part by NSF grant CRII: CIF: 1755656 and in part by NSF grant CIF: 1763734.

## Footnotes

[1] Here, we adopt the standard notation $[\nu] = \{1, 2, \ldots, \nu\}$ for any positive integer $\nu$.

[2]An equivalent definition of the problem from the perspective of causal inference can be found in Appendix B.

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
