[Supplementary Material · NeurIPS_Appendix.pdf]

# Appendix

## A Setup for simulations

In this section we describe setup for simulations we conducted in this work. Notice that in contrast to the test of Tomkins et al. [33] which operates on accept/reject decisions of SB reviewers and scores provided by DB reviewers, the test we introduce in this work operates on decisions of both SB and DB reviewers. Hence, to compare the tests we need to specify (i) models of DB/SB reviewers' decisions and (ii) models of DB reviewers' scores. All simulations are run for 5000 iterations.

### A.1 Simulations in Section 3

We now provide necessary details for the simulations in Section 3.

#### A.1.1 Measurement error (Figure 2a)

For this simulation we consider the following model of SB and DB reviewers:

$$\text{DB:} \quad \log \frac{\pi_j^{(\text{db})}}{1 - \pi_j^{(\text{db})}} = \beta_0 + \beta_1 q_j^* \tag{4a}$$

$$\text{SB:} \quad \log \frac{\pi_{ij}^{(\text{sb})}}{1 - \pi_{ij}^{(\text{sb})}} = \beta_0 + \beta_1 q_j^* + \beta_2 w_j, \tag{4b}$$

that is, model (1) is correct and reviews given by the same reviewer for different papers are independent. Notice that under this model all reviewers are identical and hence issues with the setup do not manifest in this case.

We set $m = 2n = 1000$ and $\mu = \lambda = 2$. At each iteration we independently sample true scores of papers $q_j^*, j \in [n]$, from uniform distribution $\mathcal{U}[-2, 2]$ and assume that mean scores by two DB reviewers assigned to a paper $j \in [n]$ estimates true score $q_j^*$ with some Gaussian noise ($\sigma = 0.7$). We then sample values of $w_j, j \in [n]$, such that correlation between $q^*$ and $w$ equals $\varphi$ for values of $\varphi$ between 0 and 0.5. To this end, we let each paper $j \in [n]$ with the score $q_j^* < 0$ have $w_j = 1$ with probability $0.5 - \gamma$ and $w_j = -1$ otherwise. Similarly, each paper $j \in [n]$ with the score $q_j^* \geq 0$ has $w_j = 1$ with probability $0.5 + \gamma$ and $w_j = -1$ otherwise. We then vary the value of $\gamma \in (0, 0.5)$ to achieve the necessary correlation. Finally, using models (4a) and (4b) with $\beta_0 = 1$, $\beta_1 = 2$ and $\beta_2 = 0$ (no bias condition) we sample decisions of SB and DB reviewers and run the DISAGREEMENT test and the test used by Tomkins et al. [33], setting the significance level to be $\alpha = 0.05$. We then compute a Type-I error as a fraction of iterations in which the null hypothesis ($\beta_2 = 0$) was rejected.

#### A.1.2 Model mismatch (Figure 2b)

For this simulation we consider a violation of model (1) and the following model of SB and DB reviewers with $\beta_2 = 0$ (no-bias condition):

$$\text{DB:} \quad \log \frac{\pi_j^{(\text{db})}}{1 - \pi_j^{(\text{db})}} = \beta_0 + \beta_1 (q_j^*)^3$$

$$\text{SB:} \quad \log \frac{\pi_{ij}^{(\text{sb})}}{1 - \pi_{ij}^{(\text{sb})}} = \beta_0 + \beta_1 (q_j^*)^3 + \beta_2 w_j.$$

To abstract out the effect of measurement error, in this section we assume that the true scores $q_j^*, j \in [n]$, are known, but the test used by Tomkins et al. [33] fits the model defined by equation (4b). Besides the change of correct model and availability of true scores $\{q_j^*, j \in [n]\}$, the simulations follow scenario we described in Appendix A.1.1.

### A.1.3 Reviewer calibration (Figure 2c)

In this simulation we model the effect of correlations introduced by reviewer calibration. More concretely, we construct a model of reviewer calibration under which the test by Tomkins et al. [33] fails to control for the Type-I error rate. In this section we assume that true scores of submissions are proportional to the clarity of the writing. We then sample clarity scores $\zeta_j, j \in [n]$, from uniform distribution $\mathcal{U}[-1, 1]$ and define $q_j^* = \zeta_j$ for each $j \in [n]$. Eventually, we consider the following model of reviewer. For each $i \in [m]$ and for each $j \in [n]$:

$$\text{DB:} \quad \pi_{ij}^{(db)} = \pi_j^{(db)} + \ell_i \times \mathbb{I}\left[\zeta_j < 0.5\right]$$
$$\text{SB:} \quad \pi_{ij}^{(sb)} = \pi_j^{(sb)} + \ell_i \times \mathbb{I}\left[\zeta_j < 0.5\right],$$

where $\ell_i$ is reviewers' leniency which equals $0.4$ with probability $0.5$ and $-0.4$ otherwise and $\pi_j^{(db)}, \pi_j^{(sb)}$ are defined by equations (4a) and (4b) with $\beta_0 = 0, \beta_1 = 0.25$ and $\beta_2 = 0$ (no bias condition). Parameters are selected to ensure that $0 \leq \pi_{ij}^{(db)}, \pi_{ij}^{(sb)} \leq 1$.

In words, the above model says that for papers with high quality of writing ($\zeta > 0.5$) reviewers understand their content well and follow models (4a) and (4b) exactly, but for papers with lower writing quality their leniency parameter influences their decision. Notice that under this model it is natural to expect that estimates of the true scores provided by DB reviewers are also influenced by their leniency and hence are noisy. However, to isolate the effect of reviewer identity we assume that the test used by Tomkins et al. [33] knows true scores $q_j^*, j \in [n]$, exactly. Additionally, notice that marginally each reviewer follows the model defined by equations (4a) and (4b), and hence when $\mu = 1$, the test by Tomkins et al. [33] has control over the Type-I error for any correlation between $q^*$ and $w$.

In this section we consider an extreme pattern of correlations between $q^*$ and $w$. Concretely, we assume that for any paper $j \in [n]$, we have $w_j = 1$ if and only if $q_j^* > 0.5$ and $w_j = -1$ otherwise. Notice that in practice such strong dependence is unlikely to happen, but we underscore that in practice the test by Tomkins et al. [33] also does not have access to noiseless true scores which will cause measurement errors and hence will exacerbate the issue.

We then perform simulations as discussed above having $n = 1000$ and $\lambda = 1$ fixed and varying the number of papers per reviewer and using the modification of the Wald test with factor variable for each reviewer added (reviewer-depedent intercept).

### A.1.4 Non-blind bidding (Figure 2d)

We consider a setting with $n = 1000, m = 2000, \lambda = \mu = 1$ and consider a property of interest "paper has a famous author". Suppose that during the bidding procedure each reviewer $i \in [m]$ gives a score $b_{ij} \in \{-1, 0, 1\}$ to each paper $j \in [n]$, where $b_{ij} = 1$ means that reviewer wants to review the paper, $b_{ij} = -1$ means that reviewer does not want to review the paper and $b_{ij} = 0$ is an intermediate between $b_{ij} = 1$ and $b_{ij} = -1$. Given the bids, the assignment is computed maximizing the total sum of the bids. Namely, for all $(i, j) \in [m] \times [n]$ let a binary indicator $A_{ij}$ equal 1 if reviewer $i$ is assigned to paper $j$ and 0 otherwise and let $\mathcal{R}_{SB} \subset [m]$ be the set of reviewers allocated to SB condition. Then the assignment of SB reviewers to papers is computed maximizing the following objective subject to the standard $(\lambda, \mu)$-load constraints.

$$\sum_{i \in \mathcal{R}_{SB}} \sum_{j \in [n]} A_{ij} b_{ij}.$$

The same objective is used to assign DB reviewers to papers. Next, we suppose that for each paper $j \in [n]$ there is a true score $q_j^* \in [0, 0.9]$ and that all reviewers belong to one of the following personality types:

- **Type A:** Lenient reviewers who accept each paper $j \in [n]$ assigned to them with probability $q_j^* + 0.1$ and want to read papers from top authors. If bidding is blind, they do not have any information about author identity and bid 0 on each paper, but if bidding is non-blind, then for each paper $j \in \mathcal{J}$ reviewer $i$ of type A places a bid $b_{ij} = 1$ and for each paper $j \in \overline{\mathcal{J}}$ she/he places a bid $b_{ij} = -1$.

- **Type B:** Accurate reviewers who accept each paper $j \in [n]$ assigned to them with probability $q_j^*$ and do not mind reviewing any paper. Independent of whether bidding is blind or not, reviewer $i$ of type B places a bid $b_{ij} = 0$ on each paper $j \in [n]$.

Notice that evaluations of reviewers of both types are unbiased — the probability of acceptance is not determined by author identities. The type of each reviewer is determined independently: reviewer $i \in [m]$ is of type A with probability $0.3$ and of type B with probability $0.7$. Independently, each paper $j \in [n]$ belongs to $\mathcal{J}$ with probability $0.3$ and to $\overline{\mathcal{J}}$ with probability $0.7$.

Having defined the setup, in each iteration we independently sample true scores of submissions from $\mathcal{U}[0, 0.9]$ (no correlation with indicator $w$) and compute two bidding matrices: (i) when SB reviewers observe author identities during bidding and (ii) when bidding is blind for both SB and DB reviewers. For each bidding matrix we compute assignments of SB and DB reviewers to papers and pass observed decisions to the DISAGREEMENT test and the test used by Tomkins et al. [33]. For the test of Tomkins et al., we assume that true scores $q_j^*, j \in [n]$, are known exactly.

### A.1.5   Non-random assignment (Figure 2e)

In this section we construct a similarity matrix $S$ and formalize the dependence of reviewer's perception of a paper on similarity between paper and reviewer that leads to the effect demonstrated in Figure 2e. We notice that the construction we provide here is artificial and serves as a proof of concept for our claim that non-random assignment may violate some key independence assumptions of statistical tests even if it is not based on reviewers' bids. While in practice we do not expect to observe such specific similarity matrices, we can still observe some more subtle manifestations of issues caused by non-randomness of the assignment.

First, in this section we assume that assignment is performed using the TPMS algorithms [9], that is, given similarity matrix $S$ between reviewers and papers, each paper is assigned to $\lambda$ reviewers in a way that each reviewer is assigned to at most $\mu$ papers such that total sum similarity of the assignment is maximized.

Second, consider a similarity matrix $S$, defined as follows. For each reviewer $i \in [m]$ and for each paper $j \in [n]$:

$$S_{ij} = (m + 1 - i) \times (n + 1 - j). \tag{5}$$

Given that reviewers are allocated to conditions at random, similarity matrices $S_{\text{SB}}$ (SB condition) and $S_{\text{DB}}$ (DB condition) are constructed by random division of rows of $S$ into two groups of equal size and stacking them into $S_{\text{SB}}$ and $S_{\text{DB}}$ correspondingly.

Third, we assume that each reviewer $i \in [m]$ has some value of threshold $z_i$ such that if reviewer $i$ is assigned to paper $j \in [n]$ in either of setups, reviewer accepts the paper with probability $\pi_{ij}$ given by:

$$\pi_{ij} = \begin{cases} 0.9 & \text{if } S_{ij} \geq z_i \\ q_j^* & \text{if } S_{ij} < z_i, \end{cases} \tag{6}$$

where $q_j^* \in [0, 0.9]$ is a true score of paper $j$. We also assume that reviewer $i$ in DB condition returns $\pi_{ij}$ as an estimate of $q_j^*$.

Fourth, for every reviewer $i$ we set a value of threshold as follows:

$$z_i = (m + 1 - i) \times (n - [(i-1)/2]), \tag{7}$$

where $[x]$ is the integral part of $x$.

Fifth and finally, we assume that true scores $q^*$ are independently sampled from $\mathcal{U}[0, 0.9]$ and sample indicators $w$ such that they are correlated with $q^*$, fixing the value of correlation $\varphi = 0.45$. We also set $\mu = \lambda = 1$ and $m = 2n = 1000$. Now we allocate half of reviewers to SB condition and half to DB condition uniformly at random. We then compare the performance of the DISAGREEMENT test and the test by Tomkins et al. using (i) TPMS assignment algorithm and (ii) random assignment algorithm.

The intuition behind our construction of matrix $S$ in equation (5) is that for any square submatrix of $S$, the TPMS algorithm with parameters $\mu = \lambda = 1$ will compute an assignment that corresponds to the diagonal of this submatrix. Coupled with specific choice of thresholds (7), probabilities of

acceptance (6) and correlation between $q^*$ and $w$ at the level of 0.45, this choice of similarity matrix ensures that under the setup of Tomkins et al., with non-zero probability most of SB reviewers will receive papers with similarities above the corresponding threshold and most of DB reviewers will receive papers with similarities below the corresponding threshold or vice versa. Hence, the assignments will be structurally different and, as demonstrated by Figure 2e, this difference will be confused with bias by both Tomkins et al. [33] and DISAGREEMENT tests. In contrast, when the assignment is chosen uniformly at random from the set of all feasible assignments, our DISAGREEMENT test is guaranteed to control for the Type-I error rate.

### A.2  Simulations in Section 1

The simulations in Section 1 were performed under the model of reviewers in (4a) and (4b) following the setup described in Appendix A.1.1 with small differences. Instead of varying the value of correlation $\varphi$ between $q^*$ and $w$, we fix the value of $\varphi$ and vary the number of papers $n$. Moreover, we independently assign papers to the sets $\mathcal{J}$ and $\overline{\mathcal{J}}$ as follows: each paper $j$ such that $q_j^* < 0$ belongs to the set $\mathcal{J}$ with probability $0.5 - \gamma$ and otherwise belongs to the set $\overline{\mathcal{J}}$, similarly, each paper $j$ with $q_j^* > 0$ belongs to the set $\mathcal{J}$ with probability $0.5 + \gamma$ and otherwise belongs to the set $\overline{\mathcal{J}}$. The value of $\gamma$ is selected to achieve the required level of correlation $\varphi$ between $q^*$ and $w$.

- For Figure 1a we set $\varphi = 0.4$ and perform simulations under $\beta_0 = 1, \beta_1 = 2, \beta_2 = 0$ (no bias), $\lambda = 2, \mu = 1$, where true scores are sampled from $\mathcal{U}[-1, 1]$. We see that for the test used by Tomkins et al. [33] a violation of Type-I error guarantees caused by measurement error coupled with correlations (see Appendix A.1.1 for details) exacerbates as sample size grows.

- For Figure 1b we set $\varphi = 0.6$ and perform simulations under $\beta_0 = 1, \beta_1 = 2, \beta_2 = -0.35$ (bias against papers that satisfy the property), $\lambda = 2, \mu = 1$, where true scores are sampled from $\mathcal{U}[-0.5, 0.5]$. We see that in this case measurement error has strong harmful impact on the power of the test used by Tomkins et al. [33].

- For Figure 1c we set $\varphi = 0$ and additionally assume that DB reviewers estimate true scores with no noise. In this case all parametric assumptions made by Tomkins et al. [33] are satisfied. We then perform simulations under $\beta_0 = 1, \beta_1 = 2, \beta_2 = 0.35$ (bias in favour of papers that satisfy the property), $\lambda = 2, \mu = 1$, where true scores are sampled from $\mathcal{U}[-1, 1]$.

## B  Causal inference viewpoint

In this section we provide an equivalent viewpoint on the formulation of the bias testing problem we consider in this paper. Recall that a decision of reviewer $i$ for paper $j$ if this reviewer is assigned to this paper in SB setup is denoted as $Y_{ij}$ and is a Bernoulli random variable with expectation $\pi_{ij}^{(\text{sb})}$. Our ultimate goal is to see whether the indicator variable $w_j \in \{-1, 1\}$ which encodes the property satisfaction has *causal impact* on decisions of SB reviewers. To this end, we assume that for each reviewer $i \in [m]$ and for each paper $j \in [n]$ probability $\pi_{ij}^{(\text{sb})}$ can be expressed as:

$$\pi_{ij}^{(\text{sb})} = \xi(r_i, q_j, w_j), \tag{8}$$

for some unknown function $\xi$ with co-domain $[0, 1]$, where $q_j$ is an anonymized content of a paper and $r_i$ is an arbitrary complex representation of a reviewer. That is, we assume that decisions of SB reviewers are determined by the paper content, reviewer identity and, possibly, authorship information.

Similarly, we assume that the probability of reviewer $i$ recommending acceptance for paper $j$ in DB condition is given by:

$$\pi_{ij}^{(\text{db})} = \xi(r_i, q_j, 0), \tag{9}$$

where the last argument of the function $\xi$ is censored, indicating that DB reviewers do not have access to the authorship information.

In this notation, the bias testing problem (Problem 1) can be equivalently formulated as follows:

**Problem 1'** (Equivalent formulation of the bias testing problem)**.** Given significance level $\alpha \in (0, 1)$, and decisions of SB and DB reviewers that are distributed according to equations (8) and (9), the goal

is to test the following hypotheses:

$$H_0 : \forall i \in [m] \; \forall j \in [n] \;\; \xi(r_i, q_j, 1) = \xi(r_i, q_j, 0) = \xi(r_i, q_j, -1)$$

$$H_1 : \forall i \in [m] \; \forall j \in [n] \begin{cases} \xi(r_i, q_j, 1) \geq \xi(r_i, q_j, 0) \\ \xi(r_i, q_j, -1) \leq \xi(r_i, q_j, 0) \end{cases} \tag{10}$$

where at least one inequality in the alternative hypothesis (10) is strict.

Observe that Theorem 1 ensures that the DISAGREEMENT test is guaranteed to be reliable for any (unknown) choice of function $\xi$ that falls under the definition of Problem 1'.

## C  Matching

Given assignment of papers to reviewers in both SB and DB conditions, we discuss two choices of matching algorithms depending on the relationship between parameters $\lambda$ and $\mu$. Notice that our goal is not to maximize a size of $\mathcal{T}$, but instead maximize a minimum of the number of papers from $\mathcal{J}$ and the number of papers from $\overline{\mathcal{J}}$ included in the $\mathcal{T}$, because the DISAGREEMENT test needs decisions for papers from both $\mathcal{J}$ and $\overline{\mathcal{J}}$ to maximize its power. Depending on relationship between $\lambda$ and $\mu$ we can solve this problem exactly or approximately.

Case 1 ($\lambda \geq \mu$). In this case, each paper can be matched to 1 SB reviewer and 1 DB reviewer by finding two separate maximum matchings (papers to SB reviewers and papers to DB reviewers) using the Hungarian matching algorithm. We formally present the matching procedure as Algorithm 1.

---

**Algorithm 1** Exact matching algorithm

---

**Input:** Assignments $A_{\mathrm{SB}}$, $A_{\mathrm{DB}}$ of SB and DB reviewers to papers, respectively.

1. Construct a graph $G$ that consists of 3 layers:
   - **Layer 1.** One node for each SB reviewer
   - **Layer 2.** One node for each paper
   - **Layer 3.** One node for each DB reviewer

   and add edges between reviewers and papers according to assignments $A_{\mathrm{SB}}$ and $A_{\mathrm{DB}}$. Set $\mathcal{T} = \emptyset$.
2. Using the Hungarian matching algorithm with uniform tie-breaking find matchings $\mathcal{M}_{\mathrm{SB}}$ and $\mathcal{M}_{\mathrm{DB}}$ where $\mathcal{M}_{\mathrm{SB}}$ (respectively $\mathcal{M}_{\mathrm{DB}}$) is a maximum 1-1 matching between SB (respectively DB) reviewers and papers (each reviewer is matched to at most 1 paper and each paper is matched to at most 1 reviewer).
3. Leave in graph $G$ only those edges that correspond to matched pairs in $\mathcal{M}_{\mathrm{SB}}$ and $\mathcal{M}_{\mathrm{DB}}$.
4. For any triple of (SB reviewer $i_1$, paper $j$, DB reviewer $i_2$) such that there is a path from a node that corresponds to reviewer $i_1$ to a node that corresponds to reviewer $i_2$ through a node that corresponds to paper $j$, add $t = (j, Y_{i_1 j}, X_{i_2 j}, w_j)$ to $\mathcal{T}$.
5. Return $\mathcal{T}$.

---

**Lemma 1.** *For any assignments of SB and DB referees to papers that satisfy $(\lambda, \mu)$-load constraints with $\lambda \geq \mu$, the matching procedure in Algorithm 1 is guaranteed to construct a set of tuples $\mathcal{T}$ such that for each paper $j \in [n]$ there is one tuple that corresponds to this paper.*

Case 2 ($\lambda < \mu$). In this case we cannot use the above idea, because there does not exist a matching such that each paper is matched to one SB and one DB reviewer, subject to a constraint that each reviewer is matched with at most one paper. While solving the exact optimization problem in this case might be hard, a simple greedy procedure constructs a sufficiently large matching for the DISAGREEMENT test to satisfy the non-trivial power requirement. The iterative greedy procedure in each iteration matches one paper from $\overline{\mathcal{J}}$ and one paper from $\mathcal{J}$ to 1 SB and 1 DB reviewer and removes those reviewers from subsequent interations to maintain the constraint that each reviewer contributes at most one decision to the set $\mathcal{T}$. We formally introduce the greedy procedure as Algorithm 2.

**Lemma 2.** *For any assignments of SB and DB referees to papers that satisfy $(\lambda, \mu)$-load constraints, the matching procedure in Algorithm 2 is guaranteed to construct a set of tuples $\mathcal{T}$ that for large*

*enough* $\min\{|\mathcal{J}|, |\overline{\mathcal{J}}|\}$ *contains at least* $c\min\{|\mathcal{J}|, |\overline{\mathcal{J}}|\}$ *tuples corresponding to papers from* $\mathcal{J}$ *and at least* $c\min\{|\mathcal{J}|, |\overline{\mathcal{J}}|\}$ *tuples corresponding to papers from* $\overline{\mathcal{J}}$, *where* $c$ *is a constant that may depend only on* $\lambda$ *and* $\mu$.

**Remark.** 1. If the set $\mathcal{T}$ constructed by the Algorithm 1 is such that there exist reviewers who do not contribute any of their decisions to this set, then one can run Algorithm 2 on assignments of these reviewers to papers and obtain the set $\mathcal{T}'$. Next, consider the updated set $\mathcal{T}^* = \mathcal{T} \cup \mathcal{T}'$ and observe that each reviewer contributes at most one decision to this set.

2. By construction both matching algorithms introduced in this section include at most one decision per reviewer in a set of tuples $\mathcal{T}$.

---

**Algorithm 2** Greedy matching algorithm

---

**Input:** Assignments $A_{\text{SB}}$, $A_{\text{DB}}$ of SB and DB reviewers to papers, respectively.

1. Construct a graph $G$ that cosists of 3 layers:
   - **Layer 1.** One node for each SB reviewer
   - **Layer 2.** One node for each paper
   - **Layer 3.** One node for each DB reviewer

   and add edges between reviewers and papers according to assignments $A_{\text{SB}}$ and $A_{\text{DB}}$. Set $\mathcal{T} = \emptyset$.

2. Find a triple (SB reviewer $i_1$, paper $j \in \mathcal{J}$, DB reviewer $i_2$) such that there is a path in graph $G$ from a node corresponding to SB reviewer to a node corresponding to DB reviewer through a node corresponding to a paper. If there are many such triples, break ties uniformly at random. If such a triple exists, define $t_1 = (j, Y_{i_1 j}, X_{i_2 j}, w_j)$, otherwise set $t_1 = \emptyset$.

3. Find a triple (SB reviewer $i_1' \neq i_1$, paper $j' \in \overline{\mathcal{J}}$, DB reviewer $i_2' \neq i_2$) such that there is a path in graph $G$ from a node corresponding to the SB reviewer to a node corresponding to the DB reviewer through a node corresponding to the paper. If there are many such triples, break ties uniformly at random. If such a triple exists, define $t_2 = (j', Y_{i_1' j'}, X_{i_2' j'}, w_{j'})$, otherwise set $t_2 = \emptyset$.

4. Update $\mathcal{T} = \mathcal{T} \cup \{t_1, t_2\}$. If both $t_1$ and $t_2$ are empty, return $\mathcal{T}$. Otherwise delete reviewers $i_1, i_1', i_2, i_2'$ from the graph $G$ together with the corresponding edges and go to Step 2.

---

Overall, let $\mathcal{A}$ denote a procedure that takes assignments $A_{\text{SB}}$ and $A_{\text{DB}}$ as input and depending on the relationship between $\lambda$ and $\mu$ calls Algorithm 1 or Algorithm 2 to construct the set $\mathcal{T}$.

We give proofs of lemmas in Appendix E.

# D  Proofs of main results

In this section we give proofs of our main results.

## D.1  Proof of Theorem 1

The proof of Theorem 1 consists of two parts. First, we show that the DISAGREEMENT test controls for the Type-I error rate. Second, we show that it also satisfies the requirement of non-trivial power. To abstract out the impact of the non-random assignment (issue e discussed in Section 3), in the proof below we assume that the assignment of papers to reviewers is performed at random. Before we delve into the proof, let us introduce a construction that we will use in this section.

Consider any set of triples $\mathcal{C}$ such that (i) each triple $c \in \mathcal{C}$ is of the form $(j, i_1, i_2)$ (one paper and two reviewers) and (ii) each reviewer $i \in [m]$ appears in at most one triple. Let $\mathbb{C}$ denote a collection of all such sets of triples. Then any set of tuples $\mathcal{T}$ passed to the DISAGREEMENT test as input corresponds to one member of $\mathbb{C}$ which is constructed as follows: for each $t \in \mathcal{T}$ let $(j_t, i_t, i_t')$ be a corresponding paper, SB reviewer and DB reviewer assigned to this paper, then $\mathcal{C} = \bigcup_{t \in \mathcal{T}} (j_t, i_t, i_t')$. Conversely, each member $\mathcal{C} \in \mathbb{C}$ gives rise to a family of sets of tuples $\mathbb{T}(\mathcal{C})$ which contains $2^{|\mathcal{C}|}$ elements and each element corresponds to a different allocation of reviewers in each triple $(j, i_1, i_2) \in \mathcal{C}$ to SB and DB conditions. For example, let $\mathcal{C} = \{(j, i_1, i_2), (j', i_1', i_2')\}$, then the family $\mathbb{T}(\mathcal{C})$ consists of four sets

of tuples:

$$\mathcal{T}_1 = \{(j, Y_{i_1 j}, X_{i_2 j}, w_j), (j', Y_{i'_1 j'}, X_{i'_2 j'}, w_{j'})\}$$
$$\mathcal{T}_2 = \{(j, Y_{i_2 j}, X_{i_1 j}, w_j), (j', Y_{i'_1 j'}, X_{i'_2 j'}, w_{j'})\}$$
$$\mathcal{T}_3 = \{(j, Y_{i_1 j}, X_{i_2 j}, w_j), (j', Y_{i'_2 j'}, X_{i'_1 j'}, w_{j'})\}$$
$$\mathcal{T}_4 = \{(j, Y_{i_2 j}, X_{i_1 j}, w_j), (j', Y_{i'_2 j'}, X_{i'_1 j'}, w_{j'})\}$$

For a moment, assume that $\lambda \geq \mu$, that is, Algorithm 1 is used to construct an input $\mathcal{T}$ for the DISAGREEMENT test. Then conditioned on the fact that the set of tuples $\mathcal{T}$ constructed by the algorithm belongs to $\mathbb{T}(\mathcal{C})$, the randomness of the allocation of reviewers to conditions, the random[3] assignment procedure used to assign reviewers to papers in each condition and randomness in the tie-breaking in the matching algorithm ensure that $\mathcal{T} \in \mathcal{U}[\mathbb{T}(\mathcal{C})]$, that is, all elements of $\mathbb{T}(\mathcal{C})$ are equally likely to be constructed and no other set of tuples can be constructed.

The same argument applies to the case when $\lambda < \mu$ and Algorithm 2 is employed. Hence, we conclude that conditioned of the fact the set of tuples $\mathcal{T}$ constructed by the meta-procedure $\mathcal{A}$ belongs to $\mathbb{T}(\mathcal{C})$, we have

$$\mathcal{T} \in \mathcal{U}[\mathbb{T}(\mathcal{C})]. \tag{11}$$

For each member $\mathcal{C} \in \mathbb{C}$, let $\mathbb{P}[\mathcal{C}]$ be probability that the meta-procedure $\mathcal{A}$ defined in Appenix C constructs a set of tuples that belongs to $\mathbb{T}(\mathcal{C})$. Notice that for some $\mathcal{C} \in \mathbb{C}$ we have $\mathbb{P}[\mathcal{C}] = 0$ which happens for example when for any triple $(j, i_1, i_2) \in \mathcal{C}$, one of the reviewers has a conflict of interests with the paper. The rest of the proof is performed conditioned on the $\mathcal{C}$ with $\mathbb{P}[\mathcal{C}] > 0$. The unconditional statement of the theorem then follows from the law of total probability.

### D.1.1 Control over Type-I error

Let $\Pi^{\mathrm{db}}$ and $\Pi^{\mathrm{sb}}(= \Pi^{\mathrm{db}})$ be arbitrary matrices that fall under the definition of the null hypothesis in Problem 1. Consider arrays $U$ and $V$ constructed in Step 2 of the DISAGREEMENT test. If any of them is empty, the test keeps the null and hence does not commit the Type-I error. Now without loss of generality assume that both $U$ and $V$ are non-empty.

The idea of the proof is to show that under the null hypothesis, entries of arrays $U$ and $V$ are mutually independent and identically distributed. Assume for a moment that it is indeed the case. Then entries of arrays $U$ and $V$ are exchangeable random variables and hence the permutation test with statistics $\tau$ defined in Step 3 of Test 1 is guaranteed to provide control over Type-I error rate for any given significance level $\alpha \in (0, 1)$ and hence the result for Type-I error control follows.

Consider any entry $u$ of array $U$. Then $u$ is a decision of SB reviewer for some paper $j_t \in \mathcal{J}$, where $t$ is a tuple that corresponds to $u$. Corresponding SB and DB reviewers disagree in their decisions, that is, $Y_{j_t} \neq X_{j_t}$. Given that our goal is to prove the result conditioned on the set $\mathcal{C}$, observation (11) guarantees that

$$Y_{j_t} | (Y_{j_t} \neq X_{j_t}) \sim \text{Bernoulli}(0.5). \tag{12}$$

Indeed, given that both $Y_{j_t}$ and $X_{j_t}$ are Bernoulli random variables, one can verify that

$$\mathbb{P}[Y_{j_t} = 1, X_{j_t} = 0] = \mathbb{P}[Y_{j_t} = 0, X_{j_t} = 1],$$

where probability is measured over the randomness in $\mathcal{T}$ as well as in reviewers decisions.

Hence, entries of $U$ are Bernoulli random variables with expectation $0.5$. Provided that each reviewer contributes at most one decision to $\mathcal{T}$, entries of $U$ are also independent. Notice that independence holds even if some paper appears in $\mathcal{T}$ multiple times (each time with different reviewers), because conditioning on disagreement in equation (12) eliminates papers' signal.

The same argument applies to entries of array $V$ and hence we have shown that under the null hypothesis entries of $V$ and $U$ are independent Bernoulli random variables with probability of success $0.5$ and thus are exchangeable.

### D.1.2 Non-trivial power

Consider any fixed choice of $\delta > 0$ and $\varepsilon > 0$ in definition of non-trivial power. The goal now is to show that there exists $n_0 > 0$ such that if $\min\{|\mathcal{J}|, |\overline{\mathcal{J}}|\} > n_0$, then for any matrices $\Pi^{\mathrm{db}}$ and $\Pi^{\mathrm{sb}}$ that satisfy alternative hypothesis in Problem 1 with margin $\delta$, the DISAGREEMENT test coupled with matching Algorithms 1 and 2 is guaranteed to reject the null with probability at least $1 - \varepsilon$. Throughout the proof we use $c$ to denote a universal constant and allow its value to change from line to line due to multiplications by some other universal constants. Lemma 1 and Lemma 2 guarantee that for large enough $n_0$, there exists a universal constant $c > 0$ such that set $\mathcal{T}$ provided to DISAGREEMENT test contains at least $cn_0$ tuples that correspond to papers from $\mathcal{J}$ and at least $cn_0$ tuples that correspond to papers from $\overline{\mathcal{J}}$. Recall that problem parameters $\lambda, \mu$ and $\alpha$ are treated as constants. For concreteness, throughout the proof we assume that the bias is in favor of papers from $\mathcal{J}$. The same argument can be repeated in case of bias against papers from $\mathcal{J}$.

Step 1. Cardinality of $U$ and $V$.

Let us first show that arrays $U$ and $V$ will with high probability contain order $n_0$ elements. To this end, recall that for tuple $t \in \mathcal{T}$ we add $Y_{j_t}$ to $U$ if (i) $w_{j_t} = 1$ and (ii) $Y_{j_t} \neq X_{j_t}$. Consider any of at least $cn_0$ tuples in set $\mathcal{T}$ which correspond to papers from $\mathcal{J}$, then equation (11) ensures that $\mathbb{P}\left[Y_{j_t} \neq X_{j_t}\right]$ is lower bounded by:

$$
\begin{aligned}
\mathbb{P}\left[Y_{j_t} \neq X_{j_t}\right] &= \frac{1}{2}\left(\pi_{i_1 j_t}^{(\mathrm{sb})}(1 - \pi_{i_2 j_t}^{(\mathrm{db})}) + \pi_{i_2 j_t}^{(\mathrm{db})}(1 - \pi_{i_1 j_t}^{(\mathrm{sb})})\right) + \frac{1}{2}\left(\pi_{i_2 j_t}^{(\mathrm{sb})}(1 - \pi_{i_1 j_t}^{(\mathrm{db})}) + \pi_{i_1 j}^{(\mathrm{db})}(1 - \pi_{i_2 j_t}^{(\mathrm{sb})})\right) \\
&\geq \frac{1}{2}\left(\pi_{i_1 j_t}^{(\mathrm{sb})}(1 - \pi_{i_2 j_t}^{(\mathrm{db})})\right) + \frac{1}{2}\left(\pi_{i_2 j_t}^{(\mathrm{sb})}(1 - \pi_{i_1 j_t}^{(\mathrm{db})})\right) \\
&\stackrel{(i)}{\geq} \frac{1}{2}\left(\delta^2 + \delta^2\right) = \delta^2,
\end{aligned}
$$

where two terms in the RHS of the first line correspond to two equiprobable allocations of reviewers $(i_1, i_2)$ to SB/DB conditions and inequality $(i)$ follows from the fact that for any reviewer $i \in [m]$ and for any paper $j \in [n]$ we have $\delta \leq \pi_{ij}^{(\mathrm{sb})} \leq 1$ and $0 \leq \pi_{ij}^{(\mathrm{db})} \leq 1 - \delta$ by the definition of non-trivial power requirement.

The same argument applies for tuples $t \in \mathcal{T}$ that correspond to papers from $\overline{\mathcal{J}}$. Hence, we conclude that for any tuple $t \in \mathcal{T}$ we are guaranteed that $Y_{j_t} \neq X_{j_t}$ with probability at least $\delta^2$.

Now notice that $|U| = \sum\limits_{t \in \mathcal{T}: w_{j_t} = 1} \mathbb{I}\left[Y_{j_t} \neq X_{j_t}\right]$ and hence $\mathbb{E}\left[|U|\right] \geq cn_0 \delta^2$. Applying Hoeffding's inequality, we can also derive that for large enough $n_0$ with probability at least $1 - \frac{\varepsilon}{4}$ we have

$$|U| > cn_0 \delta^2. \tag{13}$$

The same argument applies to $V$ and hence we conclude that with probability at least $1 - \frac{\varepsilon}{2}$ we have

$$|U| > cn_0 \delta^2 \quad \text{and} \quad |V| > cn_0 \delta^2.$$

Step 2. Distribution.

Now we describe the distribution of components of $U$ and $V$. By construction, the entries of these arrays are independent, so it suffices to study a single component. Consider an entry $u$ of array $U$ and let $(j, i_1, i_2)$ be a corresponding triple from the set $\mathcal{C}$. For compactness, let $p = \pi_{i_1 j}^{(\mathrm{sb})} \in (\delta, 1]$, $q = \pi_{i_2 j}^{(\mathrm{db})} \in [0, 1-\delta)$, $\gamma_1 = p - \pi_{i_1 j}^{(\mathrm{db})}$ and $\gamma_2 = \pi_{i_2 j}^{(\mathrm{sb})} - q$, where $\gamma_1 > \delta$ and $\gamma_2 > \delta$ by the definition of non-trivial power requirement. Then, we can derive the following chain of bounds:

$$
\begin{aligned}
2\mathbb{P}\left[u = 1\right] - 1 &= 2\mathbb{P}\left[Y_j = 1 | Y_j \neq X_j\right] - 1 \\
&= \frac{p(1-q)}{p(1-q) + q(1-p)} + \frac{(q + \gamma_2)(1 - p + \gamma_1)}{(q + \gamma_2)(1 - p + \gamma_1) + (p - \gamma_1)(1 - q - \gamma_2)} - 1 \\
&\stackrel{(i)}{\geq} \frac{p(1-q)}{p(1-q) + q(1-p)} + \frac{(q + \delta)(1 - p + \delta)}{(q + \delta)(1 - p + \delta) + (p - \delta)(1 - q - \delta)} - 1 \\
&= \frac{1}{2}\left(\frac{p - q}{p + q - 2pq} - \frac{p - q - 2\delta}{p + q - 2pq + 2\delta(\delta + q - p)}\right)
\end{aligned}
$$

where inequality $(i)$ holds due to monotonicity of the expression over $\gamma_1$ and $\gamma_2$ and lower bounds $\gamma_1 > \delta$, $\gamma_2 > \delta$.

Optimizing the last expression over $p \in (\delta, 1]$ and $q \in [0, 1 - \delta)$, we obtain

$$2\mathbb{P}\left[u = 1\right] - 1 \geq \frac{\delta^2}{\delta^2 + (1-\delta)^2},$$

and hence $\mathbb{P}\left[u = 1\right] \geq \frac{1}{2} + \frac{1}{2}\frac{\delta^2}{\delta^2+(1-\delta)^2} = \frac{1}{2} + \gamma$. Similarly, we can show that $\mathbb{P}\left[v = 1\right] \leq \frac{1}{2} - \frac{1}{2}\frac{\delta^2}{\delta^2+(1-\delta)^2} = \frac{1}{2} - \gamma$, where $\gamma > 0$ is a constant that depends on $\delta$.

Step 3. Permutation.

At this point we are guaranteed that vectors $V$ and $U$ constructed in Step 2 of the DISAGREEMENT test, with probability $1 - \frac{\varepsilon}{2}$, contain at least $cn_0\delta^2$ elements and their entries are independent Bernoulli random variables. Moreover, the entries of $U$ have expectations larger than $1/2 + \gamma$ and entries of $V$ have expectations smaller than $1/2 - \gamma$, where $\gamma$ is independent of $n_0$.

Conditioned on $\min\{|V|, |U|\} > cn_0\delta^2$, notice that as $n_0$ grows, the permutation test for exchange-ablility of entries of $V$ and $U$ has power growing to 1. Hence, there exists $n_0^*$ such that if $n_0 > n_0^*$, then the permutation test rejects the null with probability at least $1 - \frac{\varepsilon}{2}$.

Finally, taking union bound over (i) probability that either of $U$ and $V$ has cardinality smaller than $cn_0\delta^2$ and (ii) probability that the permutation test fails to reject the null given $\min\{|V|, |U|\} > cn_0\delta^2$, we deduce that conditioned on $\mathcal{C}$, the requirement of non-trivial power is satisfied. It now remains to notice that the established fact holds for any $\mathcal{C}$ with $\mathbb{P}\left[\mathcal{C}\right] > 0$ and hence and hence Theorem 1 holds.

### D.2 Proof of Theorem 2

The proof consists of two parts. We first show that the DISAGREEMENT test guarantees a control over the Type-I error and satisfy the requirement of non-trivial power when only intercepts are allowed to be different. We then conclude the proof with impossibility result in case not only intercepts, but also coefficients in front of $q^*$ are allowed to be different.

#### D.2.1 Positive result

Again, the proof is presented in two parts: control over Type-I error and non-trivial power. The conceptual difference from the proof of the corresponding result for absolute bias problem is that now the parametric relationships (3) allow us to avoid the conditioning on $\mathcal{C}$ which was necessary to prove Theorem 1.

**Control over Type-I error**

Let $\Pi^{\mathrm{db}}$ and $\Pi^{\mathrm{sb}}$ be arbitrary matrices generated from the generalized logistic model under the absence of bias ($\beta_2^{(\mathrm{sb})} = 0$). Consider arrays $U$ and $V$ constructed in Step 2 of the DISAGREEMENT test from the set of tuples $\mathcal{T}$ passed to the test by procedure $\mathcal{A}$ defined in Appendix C. If any of them is empty, the test keeps the null and hence does not commit the Type-I error. Now without loss of generality assume that both arrays $U$ and $V$ are non-empty. Following the idea of the proof of Theorem 1, we need to show that entries of arrays $U$ and $V$ are exchangeable random variables. First, the mutual independence follows from construction of the set $\mathcal{T}$. Second, using equation (3), we deduce that for any paper $j \in [n]$ and for any reviewer $i \in [m]$:

$$\log \frac{\pi_{ij}^{(\mathrm{sb})}(1 - \pi_{ij}^{(\mathrm{db})})}{\pi_{ij}^{(\mathrm{db})}(1 - \pi_{ij}^{(\mathrm{sb})})} = \beta_0^{(\mathrm{sb})} - \beta_0^{(\mathrm{db})}.$$

Noticing that $\pi_{ij}^{(\mathrm{sb})}$ and $\pi_{ij}^{(\mathrm{db})}$ under the generalized logistic model are independent of reviewer's identity, we drop index $i$ from the above equation. Now we consider any entry $u$ of array $U$ together

with a corresponding tuple $t = (j_t, Y_{j_t}, X_{j_t}, w_{j_t})$ and conclude that:

$$
\begin{aligned}
\mathbb{P}\left[u = 1\right] &= \mathbb{P}\left[Y_{j_t} = 1 | Y_{j_t} \neq X_{j_t}\right] \\
&= \frac{\pi_{j_t}^{(\mathrm{sb})}(1 - \pi_{j_t}^{(\mathrm{db})})}{\pi_{j_t}^{(\mathrm{sb})}(1 - \pi_{j_t}^{(\mathrm{db})}) + \pi_{j_t}^{(\mathrm{db})}(1 - \pi_{j_t}^{(\mathrm{sb})})} \\
&= \frac{1}{1 + \frac{\pi_{j_t}^{(\mathrm{db})}(1 - \pi_{j_t}^{(\mathrm{sb})})}{\pi_{j_t}^{(\mathrm{sb})}(1 - \pi_{j_t}^{(\mathrm{db})})}} \\
&= \frac{1}{1 + e^{-(\beta_0^{(\mathrm{sb})} - \beta_0^{(\mathrm{db})})}}.
\end{aligned}
\tag{14}
$$

Importantly, the value of the paper representation $q_j$ does not appear in equation (14), implying that entries of array $U$ are identically distributed. Applying the same argument to entries of array $V$ we deduce that entries of arrays $U$ and $V$ are exchangeable random variables and hence the permutation test with the test statistic $\tau$ defined in Step 3 of Test 1 is guaranteed to control for the Type-I error rate at any given significance level $\alpha \in (0, 1)$ which concludes the proof.

**Non-trivial power**

Consider any fixed choice of $\delta > 0$ and $\varepsilon > 0$ in the definition of non-trivial power. The goal now is to show that there exists $n_0 = n_0(\varepsilon, \delta)$ such that if $\min\{|\mathcal{J}|, |\overline{\mathcal{J}}|\} > n_0$, then for any matrices $\Pi^{\mathrm{db}}$ and $\Pi^{\mathrm{sb}}$ generated from the generalized logistic model that fall under the definition of the non-trivial power requirement, the DISAGREEMENT test coupled with matching procedure $\mathcal{A}$ is guaranteed to reject the null hypothesis with probability at least $1 - \varepsilon$. Throughout the proof we use $c$ to denote a universal constant and allow its value to change from line to line due to multiplications by some other universal constants. Recall that problem parameters $\lambda, \mu$ and $\alpha$ are treated as constants. For concreteness, throughout the proof we assume that the bias is in favor of papers from $\mathcal{J}$. The same argument can be repeated in case of bias against papers from $\mathcal{J}$.

Step 1. Cardinality of $U$ and $V$.

Consider any matrices $\Pi^{\mathrm{sb}}$ and $\Pi^{\mathrm{db}}$ generated from the generalized logistic model with $\beta_2^{(\mathrm{sb})} \neq 0$ that fall under the definition of the non-trivial power requirement with margin $\delta$. First, we notice that scores $q_j, j \in [n]$, and model coefficients are bounded in absolute value by some constant $B$, and hence using equation (3) we conclude that for all $(i, j) \in [n] \times [m]$

$$
\pi_{ij}^{(\mathrm{db})} \in (\ell, b) \quad \forall j \in [n],
\tag{15}
$$

where $0 < \ell < b < 1$ and values of $\ell$ and $b$ are determined by $B$. Now consider any tuple $t = (j_t, Y_{i_1 j_t}, X_{i_2 j_t}, w_{j_t})$ from the set of tuples $\mathcal{T}$. Then

$$
\begin{aligned}
\mathbb{P}\left[Y_{i_1 j_t} \neq X_{i_2 j_t}\right] &= \pi_{i_1 j_t}^{(\mathrm{sb})}(1 - \pi_{i_2 j_t}^{(\mathrm{db})}) + \pi_{i_2 j_t}^{(\mathrm{db})}(1 - \pi_{i_1 j_t}^{(\mathrm{sb})}) \\
&\geq \min\{\pi_{i_2 j_t}^{(\mathrm{db})}, 1 - \pi_{i_2 j_t}^{(\mathrm{db})}\}\left(\pi_{i_1 j_t}^{(\mathrm{sb})} + 1 - \pi_{i_1 j_t}^{(\mathrm{sb})}\right) \\
&= \min\{\pi_{i_2 j_t}^{(\mathrm{db})}, 1 - \pi_{i_2 j_t}^{(\mathrm{db})}\} \\
&\geq \min\{\ell, 1 - b\},
\end{aligned}
$$

where the last inequality follows from equation (15). Applying Hoeffding's inequality in the same way as we did in the proof of Theorem 1 to get the bound (13), we deduce that with probability at least $1 - \frac{\varepsilon}{2}$, cardinalities of arrays $U$ and $V$ are at least $cn_0$ for some constant $c$ that may depend on $\delta$ and $B$.

Step 2. Distribution

Under the generalized logistic model defined by equation (3), for any tuple $t = (j_t, Y_{i_1 j_t}, X_{i_2 j_t}, w_{j_t})$ that belongs to the tuple set $\mathcal{T}$ we are guaranteed that

$$\mathbb{P}\left[Y_{i_1 j_t} = 1 | Y_{i_1 j_t} \neq X_{i_2 j_t}\right] = \frac{1}{1 + e^{-(\beta_0^{(\text{sb})} - \beta_0^{(\text{db})} + \beta_2^{(\text{sb})} w_{j_t})}}.$$

Hence, we are guaranteed that entries of array $U$ and $V$ are independent Bernoulli random variables such that for any $u \in U$ we have

$$\mathbb{P}\left[u = 1\right] \geq \frac{1}{1 + e^{-(\beta_0^{(\text{sb})} - \beta_0^{(\text{db})})}} + \gamma,$$

and for any entry $v$ of $V$ we have

$$\mathbb{P}\left[v = 1\right] \leq \frac{1}{1 + e^{-(\beta_0^{(\text{sb})} - \beta_0^{(\text{db})})}} - \gamma,$$

where $\gamma > 0$ is a constant that may depend on $\delta$ and $B$.

Step 3. Permutation.

At this point we are guaranteed that vectors $V$ and $U$ constructed in Step 2 of the DISAGREEMENT test, with probability $1 - \frac{\varepsilon}{2}$, contain at least $cn_0$ elements and their entries are independent Bernoulli random variables. Moreover, the entries of $U$ have expectations larger than $\frac{1}{1 + e^{-(\beta_0^{(\text{sb})} - \beta_0^{(\text{db})})}} + \gamma$ and entries of $V$ have expectations smaller than $\frac{1}{1 + e^{-(\beta_0^{(\text{sb})} - \beta_0^{(\text{db})})}} - \gamma$, where $\gamma$ is independent of $n_0$, but depends on $\delta$ and $B$.

Conditioned on $\min\{|V|, |U|\} > cn_0$, notice that as $n_0$ grows, the permutation test for exchangeablility of entries of $V$ and $U$ has power growing to 1. Hence, there exists $n_0^*$ such that if $n_0 > n_0^*$, then the permutation test rejects the null with probability at least $1 - \frac{\varepsilon}{2}$.

Finally, taking union bound over (i) probability that either of $U$ and $V$ has cardinality smaller than $cn_0$ and (ii) probability that the permutation test fails to reject the null given $\min\{|V|, |U|\} > cn_0$, we deduce that the requirement of non-trivial power is satisfied.

### D.2.2 Negative result

Now we show that if not only intercepts, but also coefficients $\beta_1^{(\text{sb})}$ and $\beta_1^{(\text{db})}$ in model (3) are allowed to be different, then no test can satisfy the requirement of non-trivial power while having reliable control over Type-I error. The high-level idea of the proof is to construct similarity matrices $\Pi^{\text{db}}$ and $\Pi^{\text{sb}}$ that simultaneously (for different choices of $(\beta_0^{(\text{sb})}, \beta_1^{(\text{sb})})$) satisfy the null and alternative hypotheses under the model (3).

We begin our construction from specifying values of $q_j, j \in [n]$. For each paper $j \in [n]$, let

$$q_j = \begin{cases} -1 & \text{if } w_j = 1 \\ 0 & \text{if } w_j = -1. \end{cases}$$

Then $\Pi^{\text{db}}$ is generated from the model of DB reviewer (3) with $\beta_0^{(\text{db})} = 0$ and $\beta_1^{(\text{db})} = 1$. In this way, for any reviewer $i \in [m]$ and for any paper $j \in [n]$, probability of accepracne $\pi_{ij}^{(\text{db})}$ satisfies:

$$M_0: \ \log \frac{\pi_{ij}^{(\text{db})}}{1 - \pi_{ij}^{(\text{db})}} = q_j.$$

That is, for any reviewer $i \in [m]$ and for any paper $j \in [n]$ we have

$$\pi_{ij}^{(\text{db})} = \begin{cases} \frac{1}{1+e} & \text{if } w_j = 1 \\ 0.5 & \text{if } w_j = -1. \end{cases}$$

We now consider two different choices of coefficients for SB reviewers. Namely, for any reviewer $i \in [m]$ and for any paper $j \in [n]$, we consider two models – one corresponding to the null hypothesis of no bias ($M_1$) and another to the alternative ($M_2$):

$$M_1\ (\beta_0^{(\text{sb})} = 1, \beta_1^{(\text{sb})} = 1, \beta_2^{(\text{sb})} = 0): \quad \log \frac{\pi_{ij}^{(\text{sb})}}{1 - \pi_{ij}^{(\text{sb})}} = 1 + q_j,$$

$$M_2\ (\beta_0^{(\text{sb})} = 3/2, \beta_1^{(\text{sb})} = 2, \beta_2^{(\text{sb})} = 1/2): \quad \log \frac{\pi_{ij}^{(\text{sb})}}{1 - \pi_{ij}^{(\text{sb})}} = \frac{3}{2} + 2q_j + \frac{1}{2}w_j.$$

Consider a matrix $\Pi^{\text{sb}}$ whose components for each $i \in [m]$ and $j \in [n]$ are defined as follows:

$$\pi_{ij}^{(\text{sb})} = \begin{cases} 0.5 & \text{if } w_j = 1 \\ \frac{1}{1 + e^{-1}} & \text{if } w_j = -1. \end{cases}$$

Then a pair of matrices $(\Pi^{\text{db}}, \Pi^{\text{sb}})$ satisfies the null hypothesis given by models $(M_0, M_1)$ that can be described by the function $f_0 \in \mathcal{F}^*$ which is defined as follows:

$$\log \frac{f_0(\pi_{ij}^{(\text{db})})}{1 - f_0(\pi_{ij}^{(\text{db})})} = 1 + \log \frac{\pi_{ij}^{(\text{db})}}{1 - \pi_{ij}^{(\text{db})}}$$

Moreover, $\Pi^{\text{db}}$ and $\Pi^{\text{sb}}$ not only satisfy the alternative $(M_0, M_2)$, but also fall under the definition of non-trivial power with margin $\delta > 0$ for function $f_0' \in \mathcal{F}^*$ that satisfies:

$$\log \frac{f_0'(\pi_{ij}^{(\text{db})})}{1 - f_0'(\pi_{ij}^{(\text{db})})} = \frac{3}{2} + 2 \log \frac{\pi_{ij}^{(\text{db})}}{1 - \pi_{ij}^{(\text{db})}}.$$

Indeed, one can verify that for all $(i, j) \in [n] \times [m]$ we have $|\pi_{ij}^{(\text{sb})} - f_0'(\pi_{ij}^{(\text{db})})| > \delta$ for some $\delta > 0$ and that sign of $\pi_{ij}^{(\text{sb})} - f_0'(\pi_{ij}^{(\text{db})})$ is determined by whether $j \in \mathcal{J}$ or $j \in \overline{\mathcal{J}}$.

Given that matrices $\Pi^{\text{db}}$ and $\Pi^{\text{sb}}$ solely determine the distribution of reviewers' decisions, we have shown that reviewers' decisions are identically distributed under both null and alternative and hence any algorithm that operates on reviewers' decision and keeps Type-I error below $\alpha$ must have power at most $\alpha$ under alternative $(M_0, M_2)$. Noticing that model $(M_0, M_2)$ falls under conditions of the non-trivial power requirement, we conclude that no algorithm can satisfy the requirement of non-trivial power without sacrificing control over Type-I error.

# E Proof of auxiliary results

In this section we give proofs for auxiliary results stated in appendix.

## E.1 Proof of Lemma 1

Consider any assignment of papers to SB reviewers that satisfy $(\lambda, \mu)$−constraint with $\lambda > \mu$. Then pick any subset of papers $\mathcal{P} \subseteq [n]$ and denote a set of SB reviewers who are assigned to at least one paper from $\mathcal{P}$ as $\mathcal{R}_{\text{SB}}$. Then one can notice that

$$|\mathcal{R}_{\text{SB}}| \geq \frac{\lambda|\mathcal{P}|}{\mu} \geq |\mathcal{P}|,$$

and hence by Hall's theorem there exists a matching that maps each paper to one reviewer such that each reviewer is matched to at most one paper. This matching is computed in Step 2 of Algorithm 1.

The same argument applies to DB reviewers and hence, joining these two matchings, the algorithm in Step 4 constructs a set of tuples $\mathcal{T}$ where for each paper $j \in [n]$ there exists a tuple that corresponds to this paper.

## E.2  Proof of Lemma 2

Consider any assignments of papers to SB and DB reviewers that satisfy $(\lambda, \mu)-$constranints. Let $\gamma$ be a maximum integer that satisfies inequality

$$\gamma \leq \min\left\{\frac{|\mathcal{J}|}{4\mu}, \frac{|\overline{\mathcal{J}}|}{4\mu}\right\}.$$

Without loss of generality, assume that $\gamma > 1$. Given that $\mu$ and $\lambda$ are treated as constants and that we only need to proof the result for large enough $\min\{|\mathcal{J}|, |\overline{\mathcal{J}}|\}$, we ignore the cases when $\min\{|\mathcal{J}|, |\overline{\mathcal{J}}|\}$ is small.

Consider a graph $G$ before the first iteration of Steps 2 - 4 of Algorithm 2. Each paper in this graph is connected to $\lambda$ SB and $\lambda$ DB reviewers such that each reviewer is connected to at most $\mu$ papers.

Now let $(i_1, j, i_2)$ and $(i'_1, j', i'_2)$ be triples found in the first iteration of the algorithm. These triples exists provided that $\gamma > 1$. Then in Step 4 we remove reviewers $i_1, i'_1, i_2, i'_2$ and corresponding edges from graph $G$. One can see that these reviewers are connected to at most $4\mu$ papers in total and hence before the second iteration of Steps 2 - 4 graph $G$ will have at least $|\mathcal{J}| - 4\mu \geq 4\mu(\gamma - 1)$ papers from $\mathcal{J}$ and $|\overline{\mathcal{J}}| - 4\mu \geq 4\mu(\gamma - 1)$ papers from $\overline{\mathcal{J}}$ that are connected to $\lambda$ SB and $\lambda$ DB remaining reviewers and each of the remaining reviewers (there must be at least $8\lambda(\gamma - 1)$ SB and $8\lambda(\gamma - 1)$ DB reviewers) will be connected to at most $\mu$ papers.

By induction we can show that in the first $\gamma$ iterations of Steps 2 - 4 the greedy algorithm will be able to find non-empty triples in Steps 2 and 3. Hence the resulting set of tuples $\mathcal{T}$ will contain at least $\gamma$ tuples that correspond to papers from $\mathcal{J}$ and at least $\gamma$ tuples that correspond to papers from $\overline{\mathcal{J}}$. We then conclude the proof noticing that $\gamma = c \min\{|\mathcal{J}|, |\overline{\mathcal{J}}|\}$, where $c$ is a constant that depends only on $\mu$.

## Footnotes

[3] From theoretical standpoint, randomness in the assignment is equivalent to the independence of reviewers' evaluations from factors that determine the assignment if it is non-random. Hence, our proof works if the assignment procedure is non-random, but reviewers' evaluations are independent of bids, similarities, etc. For brevity, we only consider the case of the random assignment here.