[Reviews · NeurIPS 2019]

Reviewer 1



UPDATE after reading response -- Thank you to the authors for the detailed response. It addresses most of my concerns. I hope the authors do include a discussion of effect sizes as they suggest in the response, since effect sizes are perhaps the most important thing to assess for a problem like this. I now see I misunderstood the importance of the assignment mchanism's confound in experimental design, compared to simple random assignment analysis; thank you for that clarification. The paper would still be strengthened if it related the problem to how it's addressed in the causal inference and experimental design literature, but the work is still a worthwhile contribution on its own. ================== Contributions 1. A simulation illustration of possible false positive errors in a widely known analysis of a WSDM double blind peer reviewing experiment. 2. Theoretical results about the challenges in FP errors when analyzing randomized reviewing experiments. 3. A proposed approach for analyzing single versus double-blind peer review, based on paired permutation tests, that has better FP error rates. ================================== The authors examine the problem of analyzing the effects of double blind review, in comparison to single blind review. Much of this work focuses on Tomkins et al., PNAS 2017, which analyzed the results of a randomized experiment for WSDM reviewing that randomly assigned reviewers to single and double blind reviewing. The authors point out major difficulties in this sort of test, and in particular, a number of flaws in Tomkins' analysis approach, and conduct a simulation showing it could have high positive error rates (much higher than the nominal p-values). The authors main contribution, I think, is a paired permutation test for assessing the statistical significance of a possible difference in review scores between the conditions (Avg Treatment Effect), conditional on a property of the paper, such as whether the first author is female; such properties constitute important substantive properties for assessing the efficacy, fairness, etc., of double-blind review. This paper also delineates some of the major issues in peer review analysis, concluding that it is difficult to solve the assignmnt problem (which I think is, the degree of control by reviewers over which papers they review). I can't tell the significance of this work. I do like the problem. I think the case for it to be in scope is because reviewing practices are of interest to the NeurIPS community. The scholarship is very weak. The theoretical investigation of the size and power of various tests feels like there should be a very large literature from the statistics and social sciences that's rlevant -- experimental design, null hypothesis tests for GLMs, etc. The authors make a few dismissive citations about regression, but little other discussion. And there are tons of randomized field experiments in the social sciences with some similarities to SB vs. DB review. Most obviously, the famous "audit" studies in economics, where names of (fake) job applicants, in their resumes, were randomly manipulated to test for ethnic and gender bias (e.g. by Mullainathan, but there's many more?). Did all those social scientists have flawed analyses too? What's so special about the SB/DB problem that social scientists haven't addressed in other contexts? In economics they have all sorts of ways of addressing correlated error terms, for example. The originality seems a little weak. The main proposed approach for testing bias, with a permutation test, is good, but perhaps a little obvious, at least if you know what a non-parametric permutation test is? (There are no citations about nonparameric testing, AFAICT.) The better contributions, in terms of originality and quality, are the theoretical analysis of issues like correlated measurements and confounded review selection. The paper would be much stronger if it adopted clearer theoretical principles from the causal inference literature. For example, the notion of what is confounding really relates to what the idealized random assignment experiment is; I'm not sure that reviewer assignment is necessarily a confound in that light, since presumably topical expertise will still be a factor in any future reviewing organization approach; perhaps this should be more clearly discussed? I have not read, and did not read, the Tomkins paper. Going by this paper's presentation of Tomkins' analysis method, I found the arguments about its flaws convincing. Averaging the scores for only one of the conditions indeed seems egregious. The authors' approach of comparing scores of individual reviews against one another is much better grounded. I'm sold that some of the methods in this paper are better than Tomkins' approach to the data analysis, but I'm not sure that means it's a big advance -- maybe Tomkins is just weak. If PNAS did crappy reviewing of their own papers (likely), that's their problem, not NeurIPS's. (The authors might want to submit this to PNAS as a response to Tomkins. That's a whole other can of worms, though.) I don't think the permutation approach in this paper is the only way to do it -- you could also imagine a logistic regression approach, using every individual review (both SB and DB) as a data point with latent quality variables and random effects, that overcomes the major issues in Tomkins' approach. I think this would be the most popular approach in relevant areas of the quantitative social sciences. It would have an advantage over the appraoch here of reporting effect sizes, as well as p-values; effect sizes, of course, are of great interest. I personally believe the approach of only caring about p-values or Type-I error rates is insufficient and sometimes dangerous (see, for example, the ASA's statement on p-values several years ago, or a zillion review articles in the psychology methods literature on this right now). The authors do not make clear what their proposed analysis approach is; they don't really mention effect sizes, so I have to assume they are missing them, which is a weakness of the proposal. To be more generous, maybe the authors intend their approach to provide p-values alongside simple empirical estimates of the effect sizes? Unfortunately, a fleshed-out analysis method isn't really presented. In sum, there's something interesting here, but it doesn't feel ready to be published.

Reviewer 2



This paper works on an issue of the testing framework used in Tomkins et al. for evaluating reviewer bias in peer review. It points out the issue of false positives with measurement error, model mismatch, reviewer calibration, and reviewer assignment. It is shocking to observe the extent of false alarm probability. It then proceeds to develop a novel model-free algorithm that is robust to the first three potential issues. Strength * This paper is well written and clearly present the issue at question. * The proposed algorithm is simple and intuitive, with strong guarantee, and is also validated in empirical simulations. * This proposed framework is general and can be applied in other settings such as hiring and admission. * Although it is a negative result, the perspective raised in 5.2 is quite intriguing and addresses a question that I had during reading the paper, i.e., SB and DB can be different simply based on the assignment into these two groups. Weakness * All the experiments are done by simulation. * It would be useful to develop a quantitative understanding of the influence on power. Additional questions * I was curious whether the matching process in the algorithm could introduce any bias similar to paper assignment process, or whether there is anyway to examine the quality of matching. Minor presentation related suggestions: "control for false alarm probability" sounds strange to me. "guarantee" seems more appropriate at most places.

Reviewer 3



The main focus of this paper is to outline faults in a test statistic given by a previous model (PNAS, 2017) and to design a better test statistic that circumvents these problems. I think the paper is very well-written and clear. The paper's contribution is theoretical in nature. It's results are, to my non-expert judgement, original. Significance: I do think that for the machine learning community, the results are of limited significance, as no new machine learning algorithm is presented nor is a theoretical ML result obtained. My main concern with this paper is its appropriateness for NeurIPS, as it does not contain any ML results and is only concerned with better test statistics for a case that is rather peculiar: Single-Blind review. This review format is furthermore not widely adoped anyway. One may argue that the paper should be submitted to PNAS, as this is where the original paper has been published. Another point of critique is the lack of empirical evidence: does the new statistic still flag reviewer biases for the dataset considered in the original paper? Minor points: some small errors in use of articles, in my view, e.g., l.205 --> should be "the problem". L. 38-39: this is not a full sentence. Possibly missing citations (but these are much more practical): * Khang et al. 2018. A Dataset of Peer Reviews (PeerRead): Collection, Insights and NLP Applications. NAACL * Gao et al. 2019. Does My Rebuttal Matter? Insights from a Major NLP Conference. NAACL

[Author Response · NeurIPS 2019]

We thank all the reviewers for their time and effort.

[APPROPRIATENESS FOR NEURIPS] As pointed out by Reviewer 1, reviewing practices are of interest to the NeurIPS
community. Moreover, *our work is applicable beyond peer review as well, in applications such as admission or hiring*,
where the evaluation assignments are not random and where one wishes to avoid excessive evaluations for the testing.
We note that importantly, *past NeurIPS editions feature a number of works on statistical testing* (e.g., "Hypothesis
Testing in Unsupervised Domain Adaptation with Applications in Alzheimer's Disease" (2016) and "Differentially
Private Uniformly Most Powerful Tests for Binomial Data" (2018) to just name a few due to space constraints here),
and we believe our work on statistical hypothesis testing aligns well with the scope of NeurIPS.

[IMPORTANCE OF THE PROBLEM] It is extremely important to note that the claim "This review format is furthermore
not widely adopted anyway" does not hold throughout most of academia. Even in computer science, most Theory
conferences use single blind. There are massive ongoing debates on SB vs. DB in many fields of academia (including
in fields such as databases which have switched to DB, but there is some push to move back to SB). Based on our
experience in this debate, those supporting SB also argue for evidence of biases *in their specific community*. Indeed, an
important roadblock in making the SB vs. DB arguments is the absence of rigorous evidence – and this is where our
work significantly contributes by designing principled procedures to answer the pressing questions in these debates.

[EFFECT SIZES] We thank Reviewer 1 for raising an important point of effect sizes. In fact, we should have mentioned
that **the test statistic of our test represents the effect size**. This is in line with the seminal work by J. Cohen ("A
power primer", 1992.) where the test statistic is suggested to estimate the effect size for the sign test. We are more than
happy to add more in-depth discussion of the effect size in the revision.

[GLMS AND OTHER MODELS] Our test makes significantly fewer assumptions than the tests based on GLMs and other
parametric models. Specifically, our test *does not rely on strong modelling assumptions* (in contrast to GLMs) and *also
holds when reviewer decisions can be completely subjective* (in contrast to models that assume a presence of "true"
[latent] qualities). Indeed, the models suggested by the reviewers are at a risk of making spurious conclusions because
of the restrictive natures of the suggested models which may not capture the highly complex decision-making process
of the human reviewers.

[ASSIGNMENT AND MATCHING] During the review period, we **have solved the open problem of designing the
experimental procedure** that leads to a reliable testing. Specifically, we have designed the experimental procedure
(allocation of reviewers to conditions, assignment and matching) that follows standard conference peer-review pipeline
(i.e., allows to perform any assignment algorithm, including TPMS) and *does not inflate Type-I error rates of the testing
procedures*. Our proposed procedure assigns reviewers to conditions and papers to reviewers jointly in a carefully
selected manner, thereby avoiding issues pertinent to the setup of Tomkins et al. All the statements in the paper hold for
this procedure, but do not require an idealized random assignment any more. If reviewers prefer, we can include it in
the final version.

[PREVIOUS WORKS] Conference peer review setup is not a fully randomized controlled trial (i.e., the reviewers are
not assigned at random) and hence *past approaches fail due to idiosyncrasies of the process*. With respect to the
specific work mentioned by Reviewer 1 (Bertrand & Mullainathan, 2004), their method assigns identities of authors to
(fabricated) documents at random. In our setup, *random assignment of author identities to real (i.e., non-fabricated)
submissions* is problematic due to various logistical and ethical issues including reviewers guessing actual authors
thereby causing biases, not all authors/researchers agreeing to have their paper/name modified, and others — this
opens a separate can of worms which should be rigorously addressed before using it in the peer review setting. We are
definitely happy to add a discussion of this and other relevant papers (including those mentioned by Reviewer 3) in the
revision.

[PERMUTATION TEST] The standard way of performing the permutation test would fail to control for the Type-I
error because of the additional confounding due to quality of submissions. Our test is a careful modification of the
permutation test which provably controls for the Type-I error rate even in presence of such confoundings. In the final
version, we are happy to detail the shortcomings of the standard permutation test in our setup.

[REAL DATA EXPERIMENT] Unfortunately, the data from the Tomkins et al. experiment is not available to us. Tomkins
et al. mention in their work that releasing this data (even in an anonymized format) would make it possible to
deanonymize reviewers. Through our developed toolkit, we are happy to assist any program chairs who are interested to
conduct tests of biases in their respective research community.

[Meta-Review · NeurIPS 2019]

I am happy to recommend acceptance in light of the updated reviews, in light of the rebuttal, and the subsequent discussion among the reviewers. This paper raises and partially addresses an important issue in understanding biases in many decisions. Given that Tomkins et al. is quite influential and is an important reference in evaluating the practice of peer review, it is important to understand possible flaws in its methodology. As this paper argued, similar issues apply to many decisions where blind review can be potentially implemented.